# Future Treatment Strategies for Cancer Patients Combining Targeted Alpha Therapy with Pillars of Cancer Treatment: External Beam Radiation Therapy, Checkpoint Inhibition Immunotherapy, Cytostatic Chemotherapy, and Brachytherapy

**DOI:** 10.3390/ph17081031

**Published:** 2024-08-05

**Authors:** Ruth Christine Winter, Mariam Amghar, Anja S. Wacker, Gábor Bakos, Harun Taş, Mareike Roscher, James M. Kelly, Martina Benešová-Schäfer

**Affiliations:** 1Research Group Molecular Biology of Systemic Radiotherapy/Translational Radiotheranostics, German Cancer Research Center (DKFZ), Im Neuenheimer Feld 280, 69120 Heidelberg, Germany; ruth.winter@dkfz-heidelberg.de (R.C.W.); mariam.amghar@dkfz-heidelberg.de (M.A.); gabor.bakos01@gmail.com (G.B.); harun.tas@dkfz-heidelberg.de (H.T.); 2Department of Radiology, Molecular Imaging Innovations Institute (MI3), Weill Cornell Medicine, 413 East 69th Street, New York, NY 10021, USA; anw4010@med.cornell.edu (A.S.W.); jak2046@med.cornell.edu (J.M.K.); 3Service Unit for Radiopharmaceuticals and Preclinical Studies, German Cancer Research Center (DKFZ), Im Neuenheimer Feld 280, 69120 Heidelberg, Germany; mareike.roscher@dkfz-heidelberg.de

**Keywords:** targeted radionuclide therapy, targeted alpha therapy, external beam radiation therapy, immunotherapy, checkpoint inhibitors, cytostatic chemotherapy, brachytherapy, combination therapy, alpha-emitters, pre-clinical and clinical studies

## Abstract

Cancer is one of the most complex and challenging human diseases, with rising incidences and cancer-related deaths despite improved diagnosis and personalized treatment options. Targeted alpha therapy (TαT) offers an exciting strategy emerging for cancer treatment which has proven effective even in patients with advanced metastatic disease that has become resistant to other treatments. Yet, in many cases, more sophisticated strategies are needed to stall disease progression and overcome resistance to TαT. The combination of two or more therapies which have historically been used as stand-alone treatments is an approach that has been pursued in recent years. This review aims to provide an overview on TαT and the four main pillars of therapeutic strategies in cancer management, namely external beam radiation therapy (EBRT), immunotherapy with checkpoint inhibitors (ICI), cytostatic chemotherapy (CCT), and brachytherapy (BT), and to discuss their potential use in combination with TαT. A brief description of each therapy is followed by a review of known biological aspects and state-of-the-art treatment practices. The emphasis, however, is given to the motivation for combination with TαT as well as the pre-clinical and clinical studies conducted to date.

## 1. Introduction

In 1937, three years after Irène and Frédéric Joliot-Curie discovered that new radioelements can be produced artificially from non-radioactive material [1], researchers at the Massachusetts Institute of Technology (MIT) injected 48 rabbits with iodine-128 (I-128, τ_½_ = 25 min) retrieved after irradiation of stable iodine-127 using a radium-beryllium source [2,3]. They followed the accumulation of this short-lived beta minus- (β^−^, 93.1%) and positron- (β^+^, 6.9%) emitter over a time course of 30 min. The subsequent publication contemplated the use of radioactive isotopes of iodine in hyperplastic and neoplastic thyroids as both a diagnostic and therapeutic tool [2,4]. Later, sodium [^131^I]iodide (Na[^131^I]I) became the first U.S. Food and Drug Administration (FDA)-approved radiopharmaceutical in 1951 and can still be found in the guidelines for treatment of differentiated thyroid cancer [5,6,7].

In the time that followed, numerous radiotherapeutic agents were developed, starting from the incorporation of radioactive iodine isotopes into smaller and larger (bio)molecules. Nevertheless, targeted radionuclide therapy (TRNT) was used as a last-line resort treatment for patients that had either not responded to or progressed after the state-of-the-art treatment. The remarkable successes of [^177^Lu]Lu-DOTA-TATE (Lutathera^®^, FDA-approved in 2018, Novartis, Basel, Switzerland) in neuroendocrine tumors and [^177^Lu]Lu-PSMA-617 (Pluvicto^®^, FDA-approved in 2022, Novartis) in metastatic castration-resistant prostate cancer (mCRPC) [8,9,10,11] promoted the integration of TRNT into standard-of-care (SoC) practices for an even greater variety of cancer types and patients. Many reviews provide excellent overviews on available radiotherapeutic compounds and delivery strategies, including small molecules, peptidomimetics, labeled antibodies (and antibody fragments), nanoparticles, and pre-targeting vectors used in radioimmunotherapy [12,13,14].

Notwithstanding the diversity of radiotherapeutic sizes, targets, and pharmacokinetics, the application of these compounds to treat cancer share some common features that define TRNT. By analogy to chemotherapy or biologically targeted therapy, radiotherapeutics are administered systemically and accumulate in the tissue of interest by exploiting properties unique to tumor cells. A resulting advantage is the opportunity to treat heavily metastasized disease that is inaccessible to external-beam radiotherapy [14] or any other local intervention, like surgery. In contrast to other systemically administered therapies, the applied amount of radiotherapeutic substance is miniscule (typically in the nmol to pmol range), limiting the toxicity of TRNT to mostly radiobiological effects [15]. Biological targets can be located on the cancer cells themselves as well as in the tumor microenvironment. They include, but are not restricted to, cell surface receptors, adhesion molecules, enzymes, transporters, and ion channels. Additionally, radiotherapeutics can capitalize on enhanced permeability effects and redox potentials [16,17]. Tuning the targeting of radiotherapeutics based on tumor characteristics puts TRNT in a position to effectively meet the growing demand for personalized treatment of cancer patients [18].

Direct effects of radiation have traditionally been quantified by the extent of DNA damage caused at the target site, revealing therein a multitude of different cell death mechanisms, senescence, and other coping strategies induced by radiation damage. More recently, radiation effects beyond DNA damage have drawn attention. Examples of these effects include sub-lethal damage to other cellular components, radiation-induced bystander effect (RIBE), and the role of the immune system [19]. The extent and nature of radiobiological effects differs between types of radiation. For a long time, TRNT was restricted to the use of β^−^-emitting radionuclides. Depending on their energies, the emitted electrons have a tissue penetration depth of up to a few millimeters. The long tissue range has been considered advantageous in tumors with a heterogenous target expression because of the substantial cross-fire effect, which describes the irradiation of tumor cells that do not directly interact with the radiotherapeutic. From a radiobiological standpoint, the linear energy transfer (LET) of approx. 0.1–2 keV/µm mainly results in the formation of reactive oxygen species (ROS) through the excitation of water (and oxygen) molecules. In a subsequent step, ROS can cause oxidative damage to cellular components, including the DNA. The direct induction of DNA single strand (SSBs) and double strand (DSBs) breaks is less common.

Targeted alpha therapy (TαT), which constitutes a significant subcategory of TRNT, has become a rapidly expanding field in more recent years. Patients currently receiving TαT have exhausted all other available treatment options and/or have become resistant to ^177^Lu- or ^90^Y-based TRNT [20,21,22]. The success of TαT in these patients is attributed to the physical properties of the α-particles, notably their increased LET of approx. 50–300 keV/µm and the decreased tissue range of 20–70 µm—which can equal a single up to a few cell diameters. Multiple reports of breaking the resistance to other types of radiation as well as hormone deprivation and chemotherapy have been published. Additionally, the shorter tissue penetration depth of α-particles allows the treatment of tumors embedded in radiosensitive tissue, and the large amounts of energy deposited are favorable even for small and micrometastases [23,24,25,26,27]. The primary impact on the cellular level is the direct induction of DSBs, which are often formed in complex clusters. This renders TαT highly efficient regardless of the cell’s oxygenation status [28]. It is assumed that a single hit of an α-particle can cause sufficient DNA damage to kill a cell [29,30]. In comparison to TRNT with β^−^ emitters and according to the current consensus, TαT exhibits an up to 5-fold increased radiobiological effectiveness (RBE), which allows a reduction in the administered activities by a factor of 1000. An early, prominent example of successful TαT is a study from 2016, in which the first two patients with advanced mCRPC were treated with [^225^Ac]Ac-PSMA-617 [25]. The first patient was not eligible for treatment with [^177^Lu]Lu-PSMA-617 due to diffuse red marrow infiltration, whereas the second patient had progressed after two cycles of [^177^Lu]Lu-PSMA-617. After four and three cycles of 100 kBq/kg [^225^Ac]Ac-PSMA-617, respectively, both patients were in radiological as well as biochemical remission as determined by a prostate-specific antigen (PSA) value below the detectable level. In addition, the first studies providing information on long-term outcome of a median five cycles of TαT with 100–150 kBq/kg [^225^Ac]Ac-DOTA-TATE in patients with metastatic gastroenteropancreatic neuroendocrine tumors were reported. [^225^Ac]Ac-DOTA-TATE treatment prolonged the progression and overall survival in end-stage patients and appeared to be safe with transient, low-grade side effects [31].

Notwithstanding the exceptional successes not only of TRNT and herein especially TαT but also other single-line treatments (external beam radiation therapy (EBRT), immunotherapy with checkpoint inhibitors (ICI), cytostatic chemotherapy (CCT), and brachytherapy (BT)), cancer remains a highly heterogenous disease that becomes increasingly difficult to cure the more it has advanced. Some degree of resistance and disease progression are observed in every therapy currently available to patients with advanced disease. Efflux pumps, downregulation of target expression or mutation of the target to prevent binding of the therapeutic agent, increased DNA repair, and hypoxia are only a few examples of the biological effects observed. Additionally, parts of the primary tumor and/or secondary sites may be inaccessible due to lack of vascularization or impenetrability to therapeutic agents. Dormancy or reversible senescence can cause relapse at a later stage, and the side effects may render the repetition of a monotherapy impossible. In clinical settings, the administration of TαT has been associated with several early treatment-emergent side effects, including xerostomia, anemia, leukopenia, thrombocytopenia, and nephrotoxicity. To overcome the limitations of standalone therapies like TαT, the field of oncology is more frequently turning toward the combination of two or more treatment options.

Figure 1 presents an overview of therapies that are either state-of-the-art or otherwise routinely employed in the treatment and management of cancer, including EBRT, immunotherapy with ICI represented by PD-1/PD-L1 blockage, CCT involving taxane-based substances, and BT. Each of these therapies can potentially be combined with TαT to complement (synergistic effect) or even enhance (additive effect) the efficacy of α-particle-emitting radionuclides. This review will provide a brief introduction into each technique and opportunities for its combination with TαT. We point out and summarize promising combination strategies for patients and give an outlook on future perspective of further combinations. With this review, we aim to ignite ideas for promising treatment strategies which could be tested in pre-clinical settings.

## 2. Comprehensive Overview of Targeted Alpha Therapy (TαT)


**Key Facts:**
α-Emitters deposit high energies over short distances, classifying them as high-LET radiation.High-LET α-radiation induces complex DNA damage, resulting in direct ionization even in hypoxic tissue.Combining TαT with other treatments can address limitations such as target downregulation, tumor heterogeneity, as well as reduce side effects of either therapy alone.


**Figure 2 pharmaceuticals-17-01031-f002:**
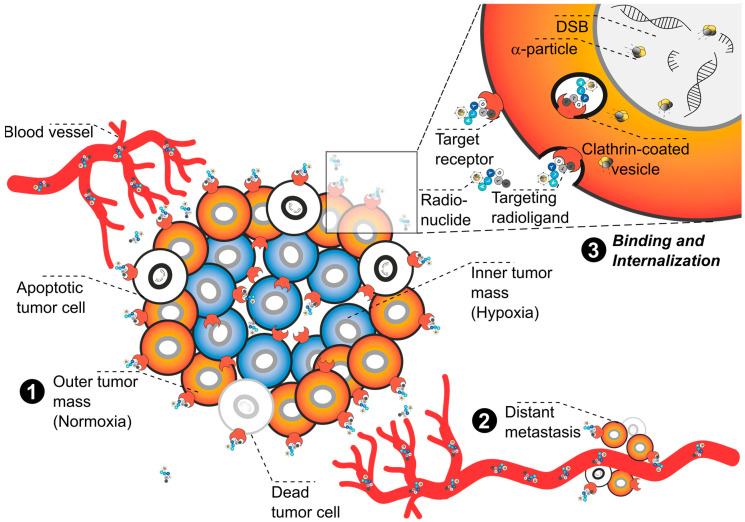
**Graphical representation of the systemic TαT**. The illustration depicts **tumor mass (1)**, **distant metastases (2)** along blood vessels, and a close-up on the **binding and internalization process** of targeting radioligands docking on target receptors **(3)**. Following intravenous injection, the radiolabeled ligand is carried to various cancer sites via the bloodstream. The targeting component enables selective binding to cancer cells through a specific target receptor. Following receptor binding, the ligand–receptor complex is internalized into the cancer cells. The high-LET radiation released upon decay damages the DNA of targeted cancer cells, mainly causing double-strand breaks (DSBs) that lead to apoptosis and other forms of cell death. Note: Due to space constraints, illustrated examples do not cover the full spectrum of possible vectors like antibodies, antibody fragments, nanoparticles, or non-conjugated α-emitters.

### 2.1. Physico-Chemical Properties and Therapeutic Implications of Alpha Particle-Emitting Radionuclides

With the exception of astatine-211 (At-211, τ_½_ = 7.21 h, 42.8% α-branching), which is a halogen and can be bound covalently to respective biovectors [32,33], all radionuclides used in TαT are radiometals. A typical pharmaceutical will therefore comprise a biovector addressing the target of interest and a chelating moiety to complex the radiometal. Most α-emitters with favorable properties for TαT have complex decay chains with several α- and β^−^/β^+^-emissions, leading to the designation of atomic nanogenerators or alpha in vivo nanogenerators [34,35]. Prominent examples include radium-223 (Ra-223, τ_½_ = 11.4 d, 4 α- and 2 β^−^-emissions), actinium-225 (Ac-225, τ_½_ = 9.9 d, 4 α- and 2 β^−^-emissions), and thorium-227 (Th-227, τ_½_ = 18.7 d, 5 α- and 2 β^−^-emissions). Other α-emitting radiometals commonly used in pre-clinical and clinical settings are lead-212/bismuth-212 (Pb-212/Bi-212, τ_½_ (Pb-212) = 10.64 h, β^−^-emission followed by α-decay with 35.9% probability), bismuth-213/polonium-213 (Bi-213/Po-213, τ_½_ (Bi-213) = 45.59 min, 2.1% α-branching and 97.9% β^−^-emission followed by α-decay), terbium-149 (Tb-149, τ_½_ = 4.12 h, 16.7% α-branching), and radium-224 (Ra-224, τ_½_ = 3.6 d, 4 α- and 2 β^−^-emissions) [36,37,38]. Emerging production and purification methods will likely increase the number of radionuclides available for TαT, as evidenced by recent studies with uranium-230 (U-230, τ_½_ = 20.2 d, 6 α- and 2 β^−^-emissions) [39,40]. Some of these radiometals form a sufficiently stable complex with the universal chelator 1,4,7,10-tetraazacyclododecane-1,4,7,10-tetraacetic acid (DOTA) for in vivo applications [38,41]. However, to achieve labeling at room temperature, which is paramount for heat-sensitive pharmaceuticals like antibodies used in radioimmunotherapy, tailored chelators have been developed for individual α-emitters. [^225^Ac]Ac^3+^, which is the largest trivalent cation in the periodic table, particularly benefits from macrocyclic chelators with increased ring size, including *N*,*N′*-bis[(6-carboxy-2-pyridil)methyl]-4,13-diaza-18-crown-6 (macropa) and its derivatives [42,43,44]. As another example, [^212^Pb]Pb^2+^ forms more stable complexes with 1,4,7,10-tetrakis(carbamoylmethyl)-1,4,7,10-tetraazacyclododecane (DOTAM/TCMC) than with DOTA [45,46]. For the formation of highly stable Th-227 conjugates, acyclic chelators like 3,2-hydroxypyridinone (3,2-HOPO) and derivatives are commonly used [47,48]. Chelation of Ra-223/224 has thus far proven challenging and with very few exceptions—macropa being the most prominent—no chelate complexes have the requisite stability for conjugation to a biovector and use in TαT. In vivo application of radium therefore remains largely limited to [^223^Ra]RaCl_2_, which mimics the binding behavior of calcium ions to areas of high bone turnover [49,50,51,52].

While in principle the molecular targets for TαT can be as versatile as those for TRNT, some additional considerations need to be made for radiometals with multiple α- and β^−^-emissions in the decay chain. The recoil energy of the first α-emission (>100 keV) is far greater than any chemical bond strength, causing release of the daughter radionuclide from the chelator [36,53,54,55,56,57]. Additionally, the daughter radionuclide may possess physical properties (i.e, charge, ionic radius) that are poorly matched with the chelating moiety. For example, the second daughter in the decay chain of Ac-225 is astatine-217 (At-217, τ_½_ = 32.63 ms), a halogen that cannot be complexed by the commonly used chelators. This is significant because redistribution of uncomplexed (grand)daughter radionuclides, which are still a source of highly toxic radiation, may otherwise contribute to long-term side effects for the patient due to unspecific accumulation in healthy organs and tissues. Internalization [58], local administration [59], or confinement of the radiopharmaceutical in a nanoparticle [60] are potentially advantageous strategies for confining ionizing radiation to the target site (Figure 2). The one exception to this paradigm is the bone-seeking agent Ra-223. Despite a half-life of 11.4 d and a decay chain comprising six consecutive α- and β^−^-decays, less than 2% of daughter radionuclides were released from the bone in mouse studies [61].

α-Emitting radionuclides decay by the emission of a helium nucleus (^4^He)—consisting of 2 protons and 2 neutrons—and a +2 charge. The kinetic energy of these α-particles ranges from 5 to 9 MeV. By contrast, β^−^- and β^+^-particles reach a maximum kinetic energy of 2–3 MeV. Upon emission, α-particles travel at approx. 5–7% of the speed of light [62,63,64,65,66], only to rapidly lose their kinetic energy in a dense track of ionization events. Depending on the initial kinetic energy, the tissue penetration depth is typically between 20 and 70 µm, which is in the range of a few cell diameters. Taken together, α-emitters are thus classified as high-LET radiation. In the Karlsruhe Chart of Nuclides [67], the vast majority of radioisotopes that decay by α-emission are located among elements of greater atomic masses (Z ≥ 82). In spite of the high energies, the low penetration depth in both organic and inorganic material presents a challenge for the direct detection of α-particles, and the necessary equipment is not readily commercially available. Hence, indirect detection of accompanying γ-emissions is often the method of choice [68]. Additionally, many radionuclides used in TαT are not stabilized by a single α-emission. The result is a complex decay chain and the transient formation of daughter nuclides, whose physical properties—including the physical half-life—differ greatly from the parent nuclide. Finally, in comparison to β^−^-emitters, for which the recoil energy rarely exceeds 25 eV, the recoil energy of an α-emission is non-negligible and, in many cases, greater than 100 keV [54]. This energy is deposited over a very short range (around 2000 Å), and the dose associated with the recoil energy therefore needs to be considered in dosimetry estimations.

### 2.2. Biological Mechanisms of Action and Cellular Response to TαT

Tumor response depends on the physical properties of the ionizing radiation and its radiobiological effects. First described for EBRT and brachytherapy, the four ‘R’s’ describe the biological factors which influence the therapy response most: repair, repopulation, reoxygenation, and redistribution. The following paragraphs will explore the extent to which these factors, most prominently repair and reoxygenation, differ for α-particles compared to β^−^-emitters, thereby highlighting the unique potential of TαT for treating cancer.

Historically, the main biological effect of ionizing radiation has been considered to be damage to DNA [69]. In general, radiation can damage DNA in one of two ways: directly or indirectly. Low-LET radiation, including β^−^-emitters and EBRT, generates ROS through the ionization of water molecules [70]. As β^−^ energy is deposited over a range of a few millimeters, spanning tens to hundreds of cells, the number of ionization events per cell is relatively low. This is due to the small mass and charge of the electron emitted from the nucleus. Water is the most abundant molecule in a cell (averaging 70% by mass) and its ionization is therefore statistically favored. The generated ROS can then ionize other molecules in the cell, such as the DNA, which typically results in the formation of SSBs and less complex DSBs. DNA damage following exposure to low-LET radiation is enhanced in oxygenated tissue and suppressed in hypoxic areas of the tumor. Especially at low doses, TRNT with β^−^-emitters and EBRT are associated with a larger proportion of sub-lethal damage compared to TαT as SSBs are easier to repair than DSBs [71]. DNA damage inflicted by α-particles, on the other hand, is characterized by more complex and often clustered strand breaks. Because of the high energies deposited over a much shorter range, the ratio of ionization events per cell is greater, and the DNA can be directly ionized by α-particles, even in hypoxic tissue. Complex DSBs are challenging to repair, leading to a higher linearity in the relationship between dose and efficacy as the influence of sublethal damage decreases [72].

Even in the absence of radiation, the DNA is subject to damage from external and internal sources. In order to preserve integrity, the cell is equipped with a multitude of sophisticated repair mechanisms, each of which is tailored to repair a specific subset of DNA damage. In the context of DSBs caused by TRNT and TαT, the most important ones are homologous recombination repair (HR), non-homologous end joining (NHEJ), and alternative end joining (Alt-EJ, which includes several pathways) [73]. HR is the most accurate DNA repair pathway but is only active in the S- and G2-phases of the cell cycle as it requires an intact homologous strand of DNA [74]. NHEJ and Alt-EJ are potentially active throughout all phases of the cell cycle but are more prone to introduce errors during repair. DNA damage repair pathways are commonly impaired, especially in aggressive cancers [75].

DNA repair typically occurs in the time frame of a few minutes to several hours, depending on the pathway, the status of the cell cycle and the severity of the damage. There are multiple entities responsible for sensing DNA damage, some of which are used as biological markers for treatment response. The most widespread example is the formation of γH2AX foci, which can be detected by specific antibodies. As a response to DNA damage, the cell halts the progression of the cell cycle, causing accumulation of cells at the G2/M checkpoint. Cell cycle arrest is another factor that is often analyzed during preclinical evaluation. In normal cells, unrepaired damage results in cell death via p53- and apoptosis executioner caspases 3- and 7-mediated pathways. In cancer cells, this pathway is often defective, and the cells continue replicating without proper DNA repair. Chromosome breaks and additional chromosomal aberrations are the result, and these ultimately cause mitotic catastrophe and the formation of micronuclei. Other, less common cell death pathways which may be triggered in response to ionizing radiation include necrosis, necroptosis, autophagy-dependent death, pyroptosis, and ferroptosis. In addition, radiation-induced senescence has been observed, where it has been argued to contribute to a favorable outcome or to be an indicator of radiation resistance [76,77].

Several studies, many of them more recent, have argued that DNA cannot be the only target of ionizing radiation. Damage to the cellular membrane as well as organelles has been claimed to be an integral part of the biological response to radiation. It is also unlikely that all cells in a tumor mass die due to direct radiation exposure. On a macroscopic level, RIBE as well as activation of the immune system have been observed. While often confined to the immediate surrounding of the irradiated cells, even systemic effects can be elicited, including what is called the abscopal effect [78].

In summary, the physical differences of α-radiation in comparison to β^−^-, γ- and X-ray radiation shifts the importance of several biological factors in characterizing the response to TαT. The high-LET α-particles cause complex, often lethal DSBs independently of oxygenation status but typically traverse only 3–5 cells. Moreover, the complex decay chain and significant recoil energies typical of α-emitters add complexity to the planning of dosimetry calculations relevant to the treatment schedule.

### 2.3. Current Advances and Innovations in TαT

In May 2013, radium-223 dichloride (^223^RaCl_2_, Xofigo^®^, Bayer, Leverkusen, Germany) became the first FDA-approved radiotherapeutic for α-therapy in mCRPC patients with symptomatic bone metastases and no known visceral metastases [49,50]. Half a year later, the European Medicine Agency (EMA) granted its authorization for the use in Europe [79]. ^223^Ra^2+^ mimics the physiological behavior of Ca^2+^ by binding hydroxyapatite and thus accumulates in areas of bone metastases associated with a high bone turnover [80]. Between 2014 and 2019, a global, prospective, observational study of Xofigo^®^ in patients with mCRPC was carried out (REASSURE, NCT02141438) to evaluate long-term safety, clinical benefit, and treatment patterns. Counting from the first cycle of Xofigo^®^ administration, median overall survival was 15.6 months. The final analysis will be reported in 2024 [81]. In addition to its use as monotherapy, the efficacy of Xofigo^®^ in combination therapies is subject to ongoing clinical trials advancing the scope beyond prostate cancer (PCa) [49]. Combinations with chemotherapy, immunotherapy, and other types of radiation therapy have been reported and will be covered in greater detail in the individual sections of this review. As of 2024, no other TαT has been approved by the FDA or the EMA, but many are being explored in ongoing clinical trials [82].

In the early days of TαT, therapeutic antibodies conjugated to short-lived radioisotopes with a single α-emission (Pb-212, Bi-212, Bi-213, and At-211) were the most frequently used targeting vectors [80,83,84,85,86]. In recent years, longer-lived radioisotopes with more complex decay chains have superseded the shorter lived α-emitters. Their physical half-life aligns more efficiently with the biological half-life of the vector and the emission of multiple ionizing daughter nuclides increases the treatment efficacy whilst lowering the required doses. The value of Ac-225 for α-radioimmunotherapy was recognized as early as 1993 [87], resulting in a phase I clinical trial of a single cycle of [^225^Ac]Ac-lintuzumab to treat 18 patients with relapsed or refractory acute myeloid leukemia [84,88,89]. Although treatment decreased tumor burden in a majority of patients, dose-limiting toxicities were observed in the highest dose groups [89,90]. In a subsequent phase I/II clinical trial of [^225^Ac]Ac-lintuzumab in patients with untreated AML (NCT02575963), the prevalence of adverse effects required further dose reduction during the study [82,91]. This pioneering work spawned a number of subsequent studies of ^225^Ac-labeled antibodies for TαT. For example, [^225^Ac]Ac-J591, a monoclonal antibody targeting PSMA for treatment of mCRPC, has just completed a phase I dose escalation study (NCT03276572) in 32 patients. Although a biochemical response (PSA decrease ≥ 50%) was only observed in half of the patients, the maximum tolerated dose was not reached in this cohort [92]. Two additional monotherapy studies are also ongoing to assess the possibility of re-treatment for patients who responded to the first dose but then progressed (NCT04576871) and to explore the efficacy of dose fractionation over multiple treatment cycles (NCT04506567) [93]. Simultaneously, [^225^Ac]Ac-J591 is under investigation for combination with [^177^Lu]Lu-PSMA I&T (NCT04886986) and pembrolizumab, an androgen receptor signaling inhibitor (NCT04946370) [94], which will be covered in more detail in Section 4. Other examples include [^225^Ac]Ac-FPI-1434, a humanized monoclonal antibody targeting the insulin-like growth factor type I receptor (NCT03746431) [95,96,97,98], and the ^227^Th-labeled anti-HER2 antibody BAY 2701439, which just completed a first-in-human study (NCT04147819) [99]. Excellent and more thorough summaries of ongoing clinical trials are available in other reviews [33,82,83].

The FDA-approval of Lutathera^®^ ([^177^Lu]Lu-DOTATATE, Novartis) and Pluvicto^®^ ([^177^Lu]Lu-PSMA-617, Novartis) for TRNT has created a significant momentum for the potential translation of rapidly distributing vectors, e.g., small molecules and peptides, to TαT. Currently, the conjugation of Ac-225 to the aforementioned vectors is one of the most frequently used strategies. The net α-energy of 28 MeV exhibited by Ac-225 strongly exceeds the therapeutic capacity of Lu-177, which possesses a maximum β^−^-energy of 0.5 MeV. Furthermore, administered activities of the ^225^Ac-labeled radiotherapeutic are significantly lower, with only one thousandth of the dose required for the corresponding ^177^Lu-labeled compound (kBq/kg vs. MBq/kg). After the initial reports in 2016 of successful treatment of mCPRC patients with [^225^Ac]Ac-PSMA-617 that no longer responded to other treatments [20,25], follow-up studies found this radiotherapeutic to be efficacious and safe as a last-line treatment in patients with mCRPC in a variety of settings [100,101]. These promising results led to an open label phase I multicycle dose escalation study seeking to recruit 60 patients with PCa, who will be subdivided in three groups according to prior treatment with chemotherapy, androgen receptor pathway inhibitors, and [^177^Lu]Lu-PSMA-617 TRNT (AcTION, NCT04597411) [102]. In parallel, a phase II clinical trial is currently evaluating the efficacy of 100 kBq/kg [^225^Ac]Ac-PSMA I&T (FPI-2265), another small molecule targeting PSMA, which will be administered to enrolled patients with mCRPC in four cycles every 8 weeks (TATCIST, NCT05219500).

Another example of small molecule Ac-225 TαT is the treatment of somatostatin receptor 2 (SSTR2)-expressing tumors. [^225^Ac]Ac-DOTATATE (RYZ101), an SSTR2 agonist, is currently being evaluated in a phase Ib/III clinical trial in patients with SSTR2-positive gastro-enteropancreatic neuroendocrine tumors that have progressed after treatment with [^177^Lu]Lu-labeled agonists and compared to SoC (ACTION-1, NCT05477576). Patients will receive up to four cycles every 8 weeks. The first part of the study is designed as a dose de-escalation study starting with 120 kBq/kg. In the first report released in May 2023, nine patients had been enrolled and no dose-limiting toxicities or serious treatment-related adverse effects were observed. Adverse effects requiring dose reduction occurred in two patients (decrease in grade 2 platelet count and grade 2 thrombocytopenia) [103]. The first dosimetry results of four patients were reported in June 2023 [104]. Aside from the comparison to SoC, [^225^Ac]Ac-DOTATATE (RYZ101) is also being investigated as an addition to SoC consisting of carboplatin, etoposide, and atezolizumab (NCT05595460). The Pb-212 analogue, [^212^Pb]Pb-DOTAMTATE (AlphaMedix), has recently completed a phase I study (NCT03466216) and is currently being evaluated in a multicenter phase II clinical trial (NCT05153772). Twenty patients were enrolled in the phase I study, none of whom had been pre-treated with TNRT. The trial was designed as a single ascending dose study, which turned into a multiple ascending dose study once response was observed. Ten patients received the highest dose (2.5 MBq/kg), with 80% of the patients presenting with a radiologic response. No serious treatment-emergent adverse effects were recorded, with only mild nausea, fatigue, and alopecia reported [105]. Moreover, four cycles of [^212^Pb]Pb-DOTAMTATE therapy were effective even in patients with progression under prior treatment of Lu-177 or Y-90 TRNT [106].

In addition to the aforementioned examples, a number of small molecules or peptides conjugated to Pb-212, Bi-213, or Ac-225 are in active or recently completed clinical trials. Examples include a phase I dose escalation study of [^212^Pb]Pb-DOTAM-GRPR1 in metastatic tumors positive for the gastrin releasing peptide receptor (NCT05283330), a phase I/IIa clinical trial using [^212^Pb]Pb-VMT-α-NET in advanced SSTR2-positive neuroendocrine tumors (NCT05636618), and a first-in-human study of [^212^Pb]VMT01 in patients with unresectable metastatic melanoma expressing the melanocortin sub-type 1 receptor (NCT05655312). A first-in-human phase I study in patients with advanced, metastatic, or recurrent solid tumors expressing the neurotensin receptor 1 is currently assessing the safety, tolerability, and dosimetry of [^225^Ac]Ac-FPI-2059 (NCT05605522). [^177^Lu]Lu- and [^90^Y]Y-labeled small molecules targeting the fibroblast activation protein-α are well established in the clinical trial landscape [107]; however, their translation to TαT is still in the preclinical phase [108,109,110,111]. Substance P is the natural ligand of the neurokinin type 1 (NK-1) receptor and can be conjugated with a metal chelator and a suitable radioisotope [112]. Phase I and II studies have been undertaken with both ^213^Bi- and ^225^Ac-labeled Substance P analogues [113]. In 2019, 20 patients presenting with recurrent glioblastoma multiforme received between one and seven cycles of [^213^Bi]Bi-DOTA-SP via local administration [114]. The median survival time from the first cycle was 7.5 months, and no severe adverse effect were reported. In 2021, Królicki et al. additionally reported the treatment of 21 glioblastoma patients with 10, 20, or 30 MBq of [^225^Ac]Ac-DOTA-SP in one–six cycles. Overall survival from start of treatment was 9.0 months, and the only grade 3 adverse events (thrombopenia) occurred in patients in the 30 MBq group [59]. A recently published study comparing the efficacy of [^213^Bi]Bi- with [^225^Ac]Ac-DOTA-SP found no significant difference in survival between the groups. However, the median overall survival from primary diagnosis is increased to 35.0 month with both TαTs, compared to standard treatment (surgery, radiotherapy, and chemotherapy) with a median overall survival of up to 9–15 months [115].

The challenges presented by recoiling daughter nuclides, the complex decay chains of the longer-lived radioisotopes as well as the inefficient complexation of Ra-223 and Ra-224 by available chelators have initiated the advancement of targeted and non-targeted nanoparticles in TαT. In theory, the encapsulation of α-particle emitting radionuclides into nanoparticles followed by intracellular delivery due to endocytosis-like mechanisms would address many of the challenges currently encountered in TαT. However, the field of nanomedicine needs to overcome certain obstacles prior to successful translation to clinical practice. The biodistribution profile remains a major concern, as it is strongly dependent on the physical properties of applied nanoparticles, e.g., size, surface, shape, or material, and often leads to accumulation in the liver, spleen, and lungs. Other challenges include the lack of production infrastructure, regulatory issues, and toxicity considerations. The vast preclinical landscape of nanoparticles and its challenges regarding clinical translation have already been thoroughly discussed in excellent review articles published to date [116,117,118].

### 2.4. Rationale for Combining TαT with Other Cancer Therapies

TαT has proven highly successful as a last-line treatment. Tumor control or reduction of tumor burden have been accomplished even in patients whose disease had become resistant to chemotherapy and other types of radiotherapy. Furthermore, TαT demonstrated good anti-tumor effect as a first-line treatment in chemotherapy-naïve metastatic hormone-sensitive PCa [119]. This is especially relevant for patients of higher age and related comorbidities who do not tolerate chemotherapy treatment. Nevertheless, there are reports of patients who either responded poorly to TαT or who even appear to show an early onset of resistance against TαT [120]. Additional investigations in these patients found mutations in genes associated with DNA damage repair [121] leading to the hypothesis that combination therapy with other pillars of cancer treatment could present a strategy in overcoming such resistances.

As with every other cancer therapy in clinical use, TαT has its limitations. Being a subcategory of TRNT, some potential shortcomings of TαT are closely tied to the general nature of TRNT. These include the limitation that downregulation or mutations in the binding site of the target of interest will reduce the accumulation of the radiotherapeutic in the cancer tissue and thus the delivered dose. Lack of vascularization or tumor heterogeneity can lead to non-homogeneous irradiation of the cancer tissue with some parts receiving sublethal doses. Especially for TαT, the cost and lack of widespread availability of the α-emitters need to be considered [32,122,123], which limits the number of patients that can be treated. Additionally, the dose calculations are highly complex, especially for radionuclides with multiple α- and β^−^-emissions in the decay chain. Redistribution of (grand)daughter radionuclides not only are difficult to quantify but also contribute to potential side effects due to irradiation of off-target tissue and excreting organs. This is significant because acutely dose-limiting toxicities have been described for [^225^Ac]Ac-PSMA-617 (salivary glands, irreversible xerostomia) [124], [^225^Ac]Ac-DOTA-TOC (chronic renal toxicity) [125], and Xofigo^®^ (substantial hematotoxicity) [126]. Moreover, long-term toxicities, which can occur up to a few years after the end of treatment, are still being assessed. Finally, although the incidence is currently lower than for other therapies, resistance to TαT and relapse shortly after treatment have been observed in some patients. The mechanism(s) of resistance, however, are not yet well understood and call for further elucidation.

The limitations of TαT can potentially be addressed by combination with other monotherapies. The choice of combination therapy will vary according to the problem at hand, but the variety of therapies available enhances the likelihood of an additive or even synergistic effect. As an example, the combination with EBRT or TRNT using β^−^-emitting radionuclides could lessen the effect of tumor heterogeneity on the therapy outcome. EBRT works independently of any target expression, either on the tumor cells themselves or in the tumor microenvironment. Heterogeneous target expression or its downregulation would therefore no longer confer an advantage to the cancer cells. As a second example, the use of TRNT with a β^−^-emitter in combination with TαT would increase the potential tumor cell kill via the cross-fire effect and more homogeneous tumor irradiation. Simultaneously, it may be possible to reduce the dose of each radiotherapeutic required, which would reduce any toxic side effects and, potentially, expand the availability of the TαT agent to a greater number of patients. On another perspective, EBRT cannot be used for treatment of oligometastasized patients; only the primary tumor and a few well-defined metastases can be irradiated. If combined with TαT, the smaller metastases would receive a simultaneous dose during EBRT treatment of larger tumor masses, circumventing the metastases’ ability to grow out and cause a relapse. In analogy, this can be applied to TRNT as β^−^-emitters cannot deposit sufficient energy in small metastases to deliver a lethal dose, yet in combination with TαT, this could be possible. Numerous practical examples exist and will be discussed in greater detail in the subsequent sections.

The overarching theme of this review is to highlight the major impact that combining TαT with different therapeutic strategies can have on the treatment of cancer patients. Instead of starting with one therapy until it is no longer effective before switching to an alternative, the combination of several strategies might leave fewer loopholes for the cancer to escape treatment while simultaneously protecting healthy tissue. Additionally, the therapy can be better tailored to the patient’s needs. For example, if a patient already has an impaired kidney function, TαT monotherapy might lead to renal failure. But if the dose could be substantially reduced by combination with EBRT, the kidneys may be spared while the effect on the tumor will be similar if not enhanced.

### 2.5. Pre-Clinical and Clinical Insights: TαT and TRNT

To date, only a handful of pre-clinical and clinical studies combining TRNT with a β^−^-emitter and TαT (or two therapeutic radionuclides in general) in so-called tandem therapies have been conducted, as outlined in Section 2.3 and Section 2.4. In the following, the focus lies on the current status of combined radionuclide therapy (Table 1).

A phase I/II clinical trial combining [^225^Ac]Ac-J591 and [^177^Lu]Lu-PSMA I&T for men with progressive mCRPC is currently ongoing at Weill Cornell Medical College (NCT04886986). The dose escalation study (phase I) enrolling 18 patients was recently completed and the results reported at the yearly meeting of the American Society of Clinical Oncology (ASCO). Up to 35 kBq/kg [^225^Ac]Ac-J591 in combination with 6.8 GBq [^177^Lu]Lu-PSMA I&T were safely administered over two cycles with 8 weeks intermission. Increasing the dose of [^225^Ac]Ac-J591 to 40 kBq/kg led to dose limiting toxicity (grade 2 or 3 thrombocytopenia) in two out of six patients after the first cycle. Seventeen of the eighteen patients had a PSA decline of any nature, and the decline was >50% in eleven patients. In addition to the therapeutic benefit of this combination approach, the study documented non-overlapping toxicities courtesy of the different pharmacokinetics of PSMA I&T (small molecule, ~1.5 kDa) and J591 (monoclonal antibody, ~150 kDa). [^177^Lu]Lu-PSMA I&T transiently accumulates in organs with PSMA expression (e.g., tumor, salivary glands, kidneys, spleen) but clears in a few days. [^225^Ac]Ac-J591, on the other hand, has a long circulation time, depositing a lower dose in the salivary glands and kidneys but a higher dose to the marrow. In the phase II, the effectiveness of the combined treatment will be assessed; study completion is expected in 2027 (NCT04886986). Taken together, both radiotherapeutics could contribute to the tumor dose while sparing other organs due to their different pharmacokinetic properties [127].

A second active clinical phase I/II trial is evaluating the safety of [^177^Lu]Lu-PSMA I&T in combination with Xofigo^®^ (AlphaBet, NCT05383079). The study is recruiting men with mCRPC which have progressed on second-generation androgen receptor agonists and were diagnosed with at least two untreated bone metastases. The hypothesis being tested is that [^177^Lu]Lu-PSMA I&T will treat metastases in soft tissue while Xofigo^®^ will simultaneously treat micrometastases in the bone, as these do not receive sufficient dose with [^177^Lu]Lu-PSMA I&T alone. Patients are receiving 7.4 GBq [^177^Lu]Lu-PSMA I&T for up to six cycles with 6 weeks intermission. A dose of 27.5 kBq/kg or 55 kBq/kg Xofigo^®^ will be administered between day 1 and day 5 of every cycle [128]. A preclinical study using the intratibial LNCaP xenograft model previously demonstrated robust anti-tumor efficacy of the combination therapy [129].

Several clinical and preclinical studies have assessed the feasibility of combining [^225^Ac]Ac-PSMA-617 and [^177^Lu]Lu-PSMA-617. In a preceding dosimetry study, 35 MBq [^177^Lu]Lu-PSMA-617 and 40 kBq [^225^Ac]Ac-PSMA-617 were calculated to deliver an equivalent dose to a subcutaneous xenograft model with C4-2 cells. Treatment of tumor-bearing mice with either 40 kBq [^225^Ac]Ac-PSMA-617 alone or a combination of 20 kBq [^225^Ac]Ac-PSMA-617 and 17 MBq [^177^Lu]Lu-PSMA-617 was found to significantly enhance overall survival of the mice, irrespective of the tumor size at the start of the treatment. By contrast, 35 MBq [^177^Lu]Lu-PSMA-617 alone had a beneficial effect on survival when treatment was started 5 weeks after inoculation (macroscopic disease) but not after 3 weeks (microscopic disease) [130]. The combination of [^177^Lu]Lu-PSMA-617 and [^225^Ac]Ac-PSMA-617 was also explored in a pilot study involving 20 patients with mCRPC who had progressed after treatment with [^177^Lu]Lu-PSMA-617 alone. All patients received one cycle of tandem therapy with activities ranging from 1.5 to 7.9 MBq [^225^Ac]Ac-PSMA-617 and 5.0 to 11.6 GBq [^177^Lu]Lu-PSMA-617. In patients with a good response, tandem therapy was followed up with [^177^Lu]Lu-PSMA-617 monotherapy to further manage the disease. Only 2 patients were diagnosed with progressive disease while 16 patients showed a PSA response, and for 13 of these patients, the PSA decrease was >50%. The study demonstrates that a response to Ac-225/Lu-177 tandem therapy is still possible even if patients have acquired resistance to [^177^Lu]Lu-PSMA monotherapy [131]. In a follow-up study, 15 TRNT-naïve mCRPC patients with visceral metastases, large tumor burden, and short PSA doubling times were subjected to two cycles of [^177^Lu]Lu-PSMA-617, at least one of which was given in combination with [^225^Ac]Ac-PSMA-617. Partial remission was observed in 8–10 patients, depending on the type of analysis (based on PSA decline or [^68^Ga]Ga-PSMA-11 imaging). Two patients experienced grade 3 anemia and another two were diagnosed with grade 1/2 xerostomia [132].

In 2019, another study of combination therapy using 3–5 MBq [^225^Ac]Ac-PSMA-617 and 3.5–7.5 GBq [^177^Lu]Lu-PSMA-617 in 30 patients with end-stage PCa was reported. In this case, the objective was to reduce the side effects of Ac-225 monotherapy, especially with respect to dose limiting xerostomia. A total of 23 patients were monitored for 8 weeks after one cycle. Although two patients reported with G3 thrombocytopenia and G2 anemia/leukocytopenia, no severe xerostomia was observed. A PSA decline of >50% was found for 10 patients, and 3 patients experienced complete remission [133]. In a follow-up, salivary gland function was analyzed more closely in 18 patients who underwent the tandem therapy and were compared to patients who received [^177^Lu]Lu-PSMA-617 or [^177^Lu]Lu-PSMA I&T monotherapy. Using dryness of mouth assessment (CTCAE v.5.0), standardized questionnaire (sXI), salivary gland scintigraphy (SGS), and [^68^Ga]Ga-PSMA-11 PET/CT as metrics, the investigators found that one cycle of Ac-225/Lu-177 tandem significantly impaired function while Lu-177 alone had only minor toxic effects. The tandem concept of co-administrating lower doses of [^225^Ac]Ac-PSMA-617 in combination with [^177^Lu]Lu-PSMA-617 can be a measure to decrease severe xerostomia [134,135].

Lastly, a pilot preclinical study has explored the combination of two α-emitters with different targeting properties. In this work, an EGFR-targeting monoclonal antibody (rituximab) was labeled with Pb-212 in a solution that also contained Ra-224. While the monoclonal antibody specifically targets (cancer) cells that overexpress EGFR, the additional bone-seeking properties of Ra-224 are able to simultaneously treat osseous lesions. The anti-tumor effect was confirmed in a metastatic breast cancer mouse model. The same combination also inhibited the growth of LNCaP spheroids [136,137].

### 2.6. Future Perspectives and Challenges

The benefits of targeted delivery of highly cytotoxic α-particles to metastasized lesions is already demonstrating efficacy in patients with advanced disease, even when resistance to other monotherapies has developed. Nevertheless, shortcomings are not to be neglected as TαT may be inefficient in targeting larger tumors or tumors with heterogeneous target expression. Furthermore, severe side-effects can greatly impact the quality of life, with the limited supply of α-emitting radionuclides crucially hindering availability to all eligible patients.

An amplified therapeutic use of TαT by combination with other treatments could effectively address these limitations and improve outcomes in cancer patients. This premise is already being evaluated by combination of TRNT and TαT, with early reports indicating that side effects can be reduced without compromising efficacy. A detailed summary of possible future combinations of TαT with other therapies is thoroughly discussed in the following sections.
pharmaceuticals-17-01031-t001_Table 1Table 1Overview of most recent and relevant publications on Targeted Alpha Therapy (TαT) as outlined in Section 2. To facilitate the orientation, the table groups references on TαT as a monotherapy first, followed by references on combination of TαT and other TRNT. Note: Only the most recent and relevant publications are included in the table to provide readers with a solid overview. More details can be found in Section 2.MethodAuthorYearStudyCancer TypeTRNT AgentMain FindingsRef.**TαT**Heynickx et al.2021ReviewmCRPCVarious PSMA radioligandsReviews salivary gland toxicity as a dose-limiting factor in PSMA-TRNT.[124]
Soldatos et al.2019ClinicalBone metastases in PCaXofigo^®^Toxicological profiling of Ra-223 for the treatment of bone metastases in PCa.[126]
Kratochwil et al.2021ClinicalSSTR tumors[^225^Ac]Ac-DOTA-TOCFive-year follow-up study on hematological and renal toxicity in patients.[125]
Sathekge et al.2023ClinicalmCRPC[^225^Ac]Ac-PSMA-617Treatment of chemotherapy-naïve patients: 20 patients (95%) any decline in PSA; 18 patients (86%) PSA decline of ≥50% [119]**TαT & TRNT**Meyer et al.2023Pre-clinicalPCa (C4-2)[^225^Ac]Ac-PSMA-617, [^177^Lu]Lu-PSMA-617Combination therapy enhanced overall survival in mice models.[130]
Scholz et al.2023Pre-clinicalPCa (LNCaP bone metastasis) Xofigo^®^, [^177^Lu]Lu-PSMA-617Demonstrated increased antitumor activity in combination therapy.[129]
Kulkarni et al.2019ClinicalmCRPC[^225^Ac]Ac-PSMA-617, [^177^Lu]Lu-PSMA-617Combination therapy reduced side effects of Ac-225 monotherapy.[133]
Khreish et al.2019ClinicalmCRPC[^225^Ac]Ac-PSMA-617, [^177^Lu]Lu-PSMA-617Demonstrated PSA response in patients progressing after Lu-177 monotherapy.[131]
Rosar et al.2021ClinicalmCRPC[^225^Ac]Ac-PSMA-617, [^177^Lu]Lu-PSMA-617Partial remission observed. Efficacy and safety established in patients with poor prognosis.[132]
Langbein et al.2019, 2022ClinicalmCRPC[^225^Ac]Ac-PSMA-617, [^177^Lu]Lu-PSMA radioligandsInvestigated salivary gland toxicity. Combination therapy significantly impaired function compared to Lu-177 alone.[134,135]
Kostos et al.2022ClinicalmCRPC[^177^Lu]Lu-PSMA I&T, Xofigo^®^Evaluating safety and efficacy. Hypothesis: combined treatment will be more effective for treating metastases.[128]
Tagawa et al.2023ClinicalmCRPC[^225^Ac]Ac-J591, [^177^Lu]Lu-PSMA I&TSafe administration with observed PSA decline > 50% in 11 patients. Non-overlapping toxicities noted.[92]

## 3. Integration of External Beam Radiation Therapy (EBRT) with Targeted Alpha Therapy (TαT)


**Key Facts:**
Combination therapy of EBRT and TαT can increase the dose delivered to the tumor while protecting healthy tissue. Its effectiveness in hypoxic tissue can overcome resistance to EBRT in poorly oxygenated tumors as well as treat oligometastatic disease.Clinical studies combining Xofigo^®^ (Ra-223) with SBRT in osteosarcoma and mCRPC demonstrate improved outcomes.


**Figure 3 pharmaceuticals-17-01031-f003:**
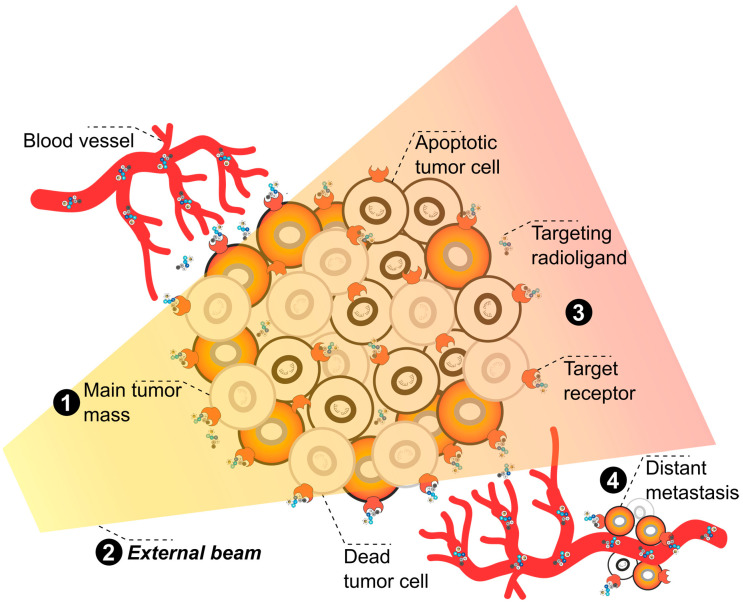
**Graphical representation of local EBRT and systemic TαT combination therapy.** The illustration depicts a **tumor mass (1)** irradiated by the **external beam** (photon irradiation) **(2)** in the form of a cropped triangle. **Targeting radioligands** docking on target receptors of the tumor cells **(3)** are addressing the main tumor mass as well as **distant metastases (4)**. Local EBRT indirectly induces DNA damage in the tumor mass. Along the radiation beam path, reactive oxygen species (ROS) are produced by ionization of water molecules. These ROS interact with the surrounding biomolecules and thereby generate DNA damage. Complex DNA damage leads to cell death. During its penetration into the tissue, the beam constantly loses energy by interaction with water (and other) molecules. As such, the penetration depth depends on the total beam energy and should be adapted to the tumor location. The beam also traverses any healthy tissue located between its entry point in the skin and the tumor as well as some tissue surrounding the tumor, which is subject to DNA damage and cell death. By contrast, the targeting compound of TαT is distributed via the blood stream and induces DNA damage primarily in those cells expressing the intended target. Moreover, systemic administration of the TαT agent will allow the targeting of distant metastases, micrometastases, and circulating tumor cells (CTCs) untreatable by EBRT. To induce the highest damage possible and to overcome radioresistance related to hypoxic conditions in the inner tumor mass, parallel/combined treatment with EBRT and TαT of the primary or local recurrent tumor mass would be favorable. The pre-treatment with EBRT could render the inner tumor mass easier accessible for TαT and, by that, enhance its efficiency. By analogy, external irradiation of the primary or recurrent tumor mass could decrease the necessary dose for TαT together with the potential TαT-related side effects. The combination of EBRT with TαT could therefore increase the dose in the tumor volume while lowering the total dose delivered to the healthy tissue by reducing the overall dose of the EBRT. Note: Due to space constraints, illustrated examples do not cover single, well-defined metastases which could also be addressed by EBRT alone.

### 3.1. Comprehensive Overview of EBRT in Cancer Treatment

EBRT is an important pillar of cancer treatment [138] administered to approximately half of all patients in Europe diagnosed with cancer [139,140]. Additionally, EBRT is also one of the most cost-effective cancer treatments, making up only about 5% of the total cost of cancer treatments [141] in spite of its extensive use. Cancer patients have been treated with radiotherapy for over a century, with the first treatment [142] recorded shortly after the discovery of X-rays by Wilhelm Conrad Roentgen in 1896 [143]. In general, radiotherapy is available globally, although, due to the specialized infrastructure and the significant initial investments for the machines, radiotherapy treatment units are far more abundant (10.1 units per million population) in high-income regions like Europe and the USA as opposed to lower-income regions like Africa or Latin America (0.01–2.6 units per million population) [144,145]. About 30% of all radiotherapy centers are located in the USA and 19% in Europe [146].

In contrast to systematic therapies like TαT, EBRT represents a local treatment, only applied in the area of the primary or recurrent tumor or to treat a few, well-defined metastases. Its efficacy is based on the induction of highly complex and difficult-to-repair DNA damage, such as clustered DSB (cDSBs). The frequency of more complex damage including cDSBs increases if the beam contains heavy particles, due to their high LET. Additionally, particle therapy benefits from a more defined irradiated area due to the particularity of the Bragg Peak, which enables the dose deposition within a sharp maximum [147]. The most commonly used sources of external beam radiation for therapy are photons and electrons, although particle therapy with protons or heavier ions like carbon is an emerging strategy which will likely gain increased clinical relevance in the future [148,149]. The drawback is related to the need for more elaborate systems, which will most likely limit the possibilities for particle therapy to high-income countries only.

### 3.2. Mechanisms Underlying Radiobiological Effects of EBRT

As described in Section 2.2, low-LET radiation (e.g., photons, electrons) primarily interacts with water molecules present in the tissue, leading to the production of oxygen radicals such as superoxide anions or hydroxyl radicals. This process, known as water radiolysis, occurs whenever radiation interacts with biological matter. These radicals, called ROS, interact with biomolecules and DNA, causing damage to accumulate in cell organelles and the DNA itself. Thus, DNA damage is mainly induced indirectly [70]. In contrast to low-LET radiation, high-LET radiation (e.g., protons, carbon ions) directly damages DNA by striking the molecule like a “bullet ball”. This causes complex DNA damage, consisting of cDSBs which are challenging to repair for the cell. Therefore, high-LET EBRT is insensitive to limiting factors, such as tumor hypoxia and cell cycle phase, which are responsible for radioresistance in many tumors. The mechanisms of DNA damage and cellular lethality are shared between high-LET EBRT and TαT. Protons and heavy ions release most of their energy at the end of the track, known as a Bragg peak. Such a specific dose distribution profile enables the majority of the dose to be delivered to the tumor while sparing the healthy surrounding tissue [150].

### 3.3. Current Advances and Innovations in EBRT

Photon therapy represents the standard of care in the field of external radiotherapy. Currently, the most advanced applications of photon therapy are 3D-conformal treatments, such as intensity-modulated radiotherapy (IMRT) [151] and image-guided radiation therapy (IGRT), enabling a tailored irradiation of tumors to, e.g., deliver a higher dose to hypoxic areas [152]. Stereotactic body radiation therapy (SBRT) has emerged as a novel treatment modality and is being evaluated in a growing number of clinical studies [153,154]. In SBRT, a biologically equivalent dose (compared to delivered doses by conventional EBRT) is delivered in a short time interval through a few high dose fractions. In one recent example, SBRT of PCa was performed in a patient with PCa using 35–36.25 Gy in five fractions. By comparison, a biologically equivalent dose of EBRT would require 40–45 fractions given a maximum dose of 2 Gy per fraction [155].

Proton therapy and carbon ion therapy are currently used to treat only specific cases due to their higher costs and limited availability. Indeed, the number of therapy centers in Europe offering carbon ion treatment is exceedingly small, namely four centers in Europe and nine in Asia as well as one center under construction in the USA (Mayo Clinic) [147,156]. Ultra-high-dose-rate external radiation therapy, named FLASH, is a promising, albeit experimental, approach to delivering EBRT. Pre-clinical studies have shown a reduction in normal tissue toxicity without compromising tumor control by delivering the conventional amount of dose in ultra-high dose rates in a short time interval (FLASH ≥ 40 Gy/s; EBRT < 0.1 Gy/s) [157]. Ongoing preclinical studies are investigating the FLASH techniques with electron, proton, and carbon ion beams [157], of which the molecular mechanisms are still unknown. Further advanced beam shaping techniques using mini-, micro-, or pencil beams also represent a highly promising treatment strategy in the future, which can be applied to photons as well as protons and heavy particles [158,159].

### 3.4. Rationale for Combining EBRT with TαT

Used in combination, TαT and EBRT could positively reinforce each other. EBRT itself is inadequate as a local treatment regime in progressive disease with multiple metastases. On the other hand, TαT may not be suitable for treating larger lesions and can cause side effects in any tissues in which the TαT agent accumulates, even if these tissues are remote from the tumor lesions. Ideally, EBRT could be used to treat the main tumor mass locally and TαT to treat the metastases systemically (Figure 3). Side effects of EBRT are mainly related to the damage induced in the healthy tissue surrounding the irradiated tumor site, as the conventional external photon beam needs to pass through this tissue to reach the tumor and a margin of error around the tumor site is needed to ensure the tumor receives the necessary dose. Combination therapy with TαT can enhance the total delivered dose in the tumor volume while sparing the surrounding healthy tissue. Significantly, the unique efficacy of TαT, even in hypoxic tissue, can overcome resistance to EBRT in poorly oxygenated tumors. First evidence of a synergistic combination of these therapies came from studies that demonstrated increased uptake of a TRNT agents ([^131^I]I-cetuximab and [^131^I]I-benzamide) after EBRT. The improved uptake was attributed to increased vascular permeability and penetrability of the tumors following fractionated EBRT [160]. By analogy, TαT treatment before and after EBRT could sensitize the tumor tissue to irradiation and enhance the effects of EBRT after therapy due to continuous delivery of radiation. Finally, the abscopal effect, stimulation of an immune response against tumor cells, is mainly associated with EBRT [78] but has been observed for TαT as well [161]. A combination of both treatments could thus enhance the likelihood of the abscopal effect and lead to greater efficacy, even in untreated tumor lesions or tumors receiving sub-lethal doses.

Further support for the potential additive effects of EBRT and TαT combination therapy can be extrapolated from preclinical studies investigating combination therapies with EBRT and β^−^-emitting TRNT agents (Table 2). These studies demonstrated that the simultaneous application of EBRT and TRNT in a fractionated scheme leads to effective tumor control [162]. A long delay between EBRT and TRNT showed a clear disadvantage. Based on these data, Hartrampf et al. proposed that a “sandwich treatment”, with TRNT ([^177^Lu]Lu-DOTA-TOC and [^177^Lu]Lu-DOTA-TATE) before and after EBRT, might be an even more effective treatment protocol [163]. Their hypothesis was based on high uptake of the TRNT agent in the pre-irradiation phase [164] coupled with EBRT-triggered enhancement of target expression that further increases TRNT agent uptake in the post-irradiation phase [165].

Cornelissen et al. combined the radionuclide indium-111 (In-111, EC, 100%, τ_½_ = 2.80 d) with the anti-γH2AX antibody coupled to the cell-penetrating peptide trans-activator of transcription (TAT) [166]. The ligand is targeting the DNA damage induced by EBRT treatment and marked by γH2AX. The combination of 60 MBq of [^111^In]In-anti-γH2AX-TAT with 10 Gy EBRT (photons) enhanced DNA damage in breast cancer MDA-MB231/H2N xenograft tumors, as evidenced by a multiplication of γH2AX phosphorylation. Combination with α-emitters might be even more beneficial given similar treatment schedules and targeting vector combinations.

### 3.5. Pre-Clinical and Clinical Insights: EBRT and TαT Synergy

To date, and due to the relatively recent emergence of TαT as a treatment option, there is a paucity of preclinical and clinical evidence for the synergistic effect of EBRT and TαT combination therapy. Nevertheless, studies with TRNT agents support this new combination treatment approach, when the physical properties of α-particles are taken into account [162]. Furthermore, two clinical studies investigating the combination of α-emitting ^223^RaCl_2_ (Xofigo^®^) with SBRT in osteosarcoma and mCRPC have served as a proof-of-concept for this approach. Xofigo^®^ was administered at 55 kBq/kg and SBRT was delivered in five fractions (5–10 Gy per fraction) over sequential days, either concurrent with TαT and/or after TαT, depending on the individual patient. In total, a maximum of six Xofigo^®^ infusions were delivered [167]. The patients who received combination therapy demonstrated a significantly improved outcome compared to those who received either monotherapy alone [167,168].

### 3.6. Future Perspectives and Challenges

Previous and current clinical studies with TRNT strongly indicate that the combination of EBRT and TαT is clinically feasible and beneficial for cancer patients. Despite offering a promising perspective, vital shortcomings remain and need to be overcome for widespread future clinical implementation. As such, protecting healthy tissue from radiation exposure persists to be a central challenge in this combinatorial approach. Hence, the dosimetry of TαT agents is a crucial variable in this regard. Utilizing a suitable diagnostic radiotracer for TαT purposes might enable superior personalized calculation of the optimal dose. However, this additional tool would increase the expenses of further prospective single photon emission computed tomography (SPECT) and positron emission tomography (PET) scans. A second challenge is the establishment of optimized dosing sequences, taking the heterogeneity of tumors within and between patients into account. FLASH radiotherapy or novel beam shaping techniques as microbeams are a potential asset in this regard, but further evaluation of these novel therapies is essential before future combinations can be considered. Last, but not least, the infrastructural requirements and costs of both therapies represent significant hurdles, which essentially might restrict the combination to a relatively small number of treatment centers throughout the world.

**Table 2 pharmaceuticals-17-01031-t002:** Overview of most recent and relevant publications on External Beam Radiation Therapy (EBRT) as monotherapy and in combined treatment with TαT (refer to Section 3). To facilitate the orientation, the table groups references on EBRT as a monotherapy first, followed by references on combination of EBRT with TRNT and TαT. Note: Only the most recent and relevant publications are included in the table to provide readers with a solid overview. More details can be found in Section 3.

Method	Authors	Year	Study	Cancer Type	EBRT Modality	TRNT Agent	Main Findings	Ref.
**EBRT**	Bilski et al.	2021	Review	Neuroendocrine tumors	Photons	NA	Role of radiotherapy in treatment of neuroendocrine cancer of the lung.	[154]
	Lin et al.	2021	Review	Various	Photons (FLASH)	NA	Ultra-high dose rate FLASH radiotherapy reduces radiation-induced damage in healthy tissue without decreasing antitumor effectiveness.	[157]
	Kundapur et al.	2024	Review	Various	Photons (Minibeam)	NA	Potential of minibeams to induce immunomodulation augmenting a direct tumoricidal effect, leading to pronounced cell kill and subsequent immune surveillance.	[156]
**EBRT & TRNT**	Oddstig et al.	2011	Pre-clinical	Small Cell Lung Carcinoma	Photons (2–8 Gy)	[^177^Lu]Lu-DOTA-TATE	Radiation induces up-regulation of somatostatin receptors in small cell lung cancer.	[165]
	Cornelissen et al.	2012	Pre-clinical	Breast cancer (MDA-MB-468)	Photons (10 Gy)	[^111^In]In-DTPA-anti-γH2AX-TAT	Amplification of DNA damage by γH2AX-targeted radiopharmaceutical and exposure to external radiation.	[166]
	Melzig et al.	2018	Pre-clinical	Head and Neck tumors (A431)	Carbon ions/Photons	[^131^I]I-cetuximab/[^131^I]I-benzamide	Combination inhibited tumor growth effectively, attributed to reduced microvascular density and decreased proliferation index.	[160]
	Hartrampf et al.	2020	Clinical	Meningioma	Photons (IMRT)	[^177^Lu]Lu-DOTATATE/-TOC	Combination of PRRT and fractionated EBRT resulted in disease stabilization in 7 of 10 patients with advanced symptomatic meningioma.	[163]
**EBRT & TαT**	Anderson et al.	2020	Clinical	Osteosarcoma	Photons (SBRT)	Xofigo	Combination therapy of Ra-223 and EBRT for metastatic osteosarcoma.	[167]
	Hasan et al.	2020	Clinical	PCa	SABR	Xofigo^®^	Randomized trial of Ra-223 and SABR versus SABR for oligometastatic PCa.	[168], NCT04037358

## 4. Integration of Immune Checkpoint Inhibitors (ICI) with Targeted Alpha Therapy (TαT)


**Key Facts**
Combining ICI with TαT can enhance immune response and tumor inflammation by inducing immunogenic cell death, overcoming resistance in heterogenous or poorly accessible tumors.Studies show that combining ICI with TRNT, such as TαT, improves survival and immune response in various cancers. Ongoing clinical trials, like those combining [^225^Ac]Ac-J591 with pembrolizumab, aim to validate these findings across different cancer types.


**Figure 4 pharmaceuticals-17-01031-f004:**
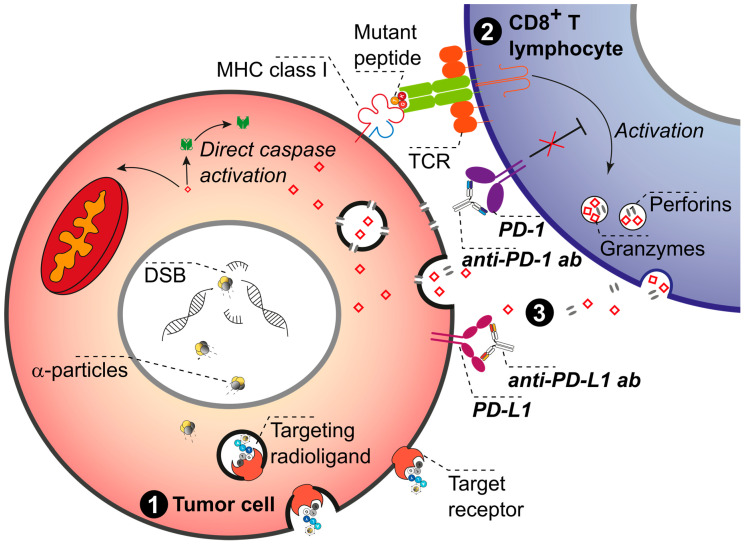
**Graphical representation of systemic immunotherapy via ICI and systemic TαT combination therapy**. The illustration depicts a single **tumor cell (red) (1)** and its interaction with a **CD8+ T lymphocyte (purple) (2)**. The immune checkpoint inhibitors **anti-PD-1 and anti-PD-L1 block the PD-1 and PD-L1 receptor–ligand complex (3)**. Such interactions facilitate cancer cell recognition and cancer cell killing by CD8+ T lymphocytes. The release of neo-antigens from tumor cells exposed to α-particle radiation enhances the recognition and killing of these cells by immune cells and activates dendritic cells, leading to improved tumor antigen presentation and immune response. At the same time, cancer cells can be irretrievably damaged by TαT, thereby killing cells which might not express PD-L1. Systemic immunotherapy with ICI activates the immune response for tumor cell killing. In parallel, systemic TαT directly kills tumor cells, even in tumors with a “cold” immune microenvironment in which immune response is suppressed. Furthermore, exposure to α-particles releases “neo-antigens” from tumor cells to stimulate both immune cell recognition and immune-mediated killing. Additionally, TαT promotes the production of many endogenous damage-associated molecular patterns (DAMPs). DAMPs activate dendritic cells, resulting in improved tumor antigen presentation, stimulation of cytotoxic T lymphocytes, and the release of cytokines and chemokines, all of which restore immune response. Therefore, the simultaneous immune checkpoint blockade and the enhancement of immune response by TαT could combine synergistically with the tumor killing effect of TαT. Note: Due to space constraints, illustrated examples do not cover the full spectrum of possible checkpoint inhibitors and focus exclusively on anti-PD-1 and anti-PD-L1 agents blocking the PD-1 and PD-L1 receptor–ligand complex.

### 4.1. Biological Basis of Immune Evasion and Restoration in ICI

The ability to elude host immunity is broadly accepted as one of the biological hallmarks of cancer. (Re)activation of the host immune system against tumors is a highly appealing strategy for treating cancer, and oncologists and immunologists have been working on new immunologic approaches for decades [169]. One reason this strategy is so appealing is that anti-tumor immune responses can, in theory, be highly specific. Blockade of T cell inhibitory molecules is one of the most promising methods for selectively enhancing immune response against cancer because it targets one of the pathways that tumor cells uniquely exploit to evade the host immune response [170]. As such, immunotherapy is perhaps the most tumor-specific treatment currently available [171].

In order to succeed, effector T cells must successfully infiltrate the tumor microenvironment (TME) and be activated upon detection of tumor antigens. Tumors have developed several strategies to disrupt this process and weaken anti-cancer immunity [172]. One strategy is to develop dense stroma that represents a physical barrier to tumor microenvironment (TME) infiltration. A second strategy is to overexpress proteins that modify T cell activation. CTLA-4, CD28, PD-1, and inducible co-stimulator (ICOS) and their ligands B7 (i.e., B7-1 or B7-2; CD80 or CD86), PD-L1, and ICOSL are members of the B7 family (within the immunoglobulin superfamily). These molecules modulate, as the so-called second signal (co-stimulation or co-inhibition), the intensity of the first signal delivered to T cells from the interaction of the T cell receptors (TCR) with the (tumor-) antigen presented in the major histocompatibility complex (MHC) [173]. The effect of the interaction is to inactivate the T cell. Disruption of the receptor–antigen binding by ICI ultimately prevents the inactivation of the T cell, enabling it to kill the tumor cell.

### 4.2. Comprehensive Overview of ICI in Cancer Treatment

Immune-checkpoint blockade is the most common strategy for restoring the host immune response for tumor cell killing. Drugs that exert their function based on this mechanism of action are known as immune checkpoint inhibitors (ICI). Anti-cytotoxic T lymphocyte antigen-4 (anti-CTLA-4), a monoclonal antibody specific for CTLA-4, the inhibitory receptor on T cells that binds the protein B7 on antigen-presenting cells, was the first ICI developed [174]. Anti-CTLA-4 immunotherapy is an FDA-approved treatment for advanced melanoma. Its mechanism of action may be to deplete T-regs, which constitutively express CTLA-4 [175]. Exhausted T cells also increase expression of CTLA-4 and are increasingly recognized as arising in cancer [176]. The programmed cell death protein 1/ligand 1 (PD-1/PD-L1) pathway is also implicated in the impaired T cell response to tumors. Antibody blockade of PD-1 or its ligand PD-L1 appears to be even more effective than anti-CTLA-4 in enhancing T cell tumor killing and preventing the progression of cancer in patients [177]. Anti-PD-1 and anti-PD-L1 antibodies also have fewer side effects than anti-CTLA-4 antibodies and are now authorized for the treatment of melanoma, lung carcinomas, renal carcinomas, bladder carcinomas, colon carcinomas, and Hodgkin lymphoma, amongst many other cancers [178]. However, these strategies do not uniformly benefit all patients. The efficacy of ICIs can be limited by factors such as the tumor microenvironment’s immunosuppressive conditions, the genetic and epigenetic landscape of the tumor, and the diversity in T cell receptor repertoire among individuals [179].

### 4.3. Current Advances and Innovations in ICI

The first antibodies against CTLA-4 which were used for in vitro functional testing and in vivo modelling consisted of hamster anti-mouse CTLA-4 and mouse anti-human CTLA-4. For the initial clinical testing, however, human-sequence anti-human CTLA-4 was used to diminish any immunogenic effects against the antibodies themselves. These anti-human CTLA-4 antibodies were developed using transgenic mice comprising a germline configuration of human immunoglobulin genes that rearrange to direct a repertoire of B cells expressing human antibodies. To date, checkpoint blockade of CTLA-4 has relied on antibodies for blocking the interaction of CTLA-4 with its ligands [180]. This inhibition must not interfere with the CD28–B7 ligand interactions required for co-stimulation of primary T cell responses [181,182]. The most prominent example of a clinically approved anti-CTLA-4 antibody is ipilimumab (Yervoy^®^, Bristol-Myers Squibb, Princeton, New Jersey, USA), which received FDA-approval in 2011 for the treatment of metastatic melanoma [183]. Ipilimumab’s approval was a milestone in cancer immunotherapy, demonstrating the potential of checkpoint inhibitors to significantly improve patient outcomes. This approach was awarded with the 2018 Nobel Prize in Physiology or Medicine for James Allison and Tasuku Honjo [184].

The first two PD-1-directed antibodies to enter the clinics, nivolumab (Opdivo^®^, Bristol Myers Squibb) and pembrolizumab (Keytruda^®^, Merck, Darmstadt, Germany), use the Fc domains of human immunoglobulin G4 (IgG4), comprising stabilizing hinge region mutations. Nivolumab was discovered in 2002 using human IgG transgenic mice, whereas pembrolizumab is the product of grafting the complementarity-determining regions of a mouse monoclonal antibody onto the homologous human antibody variable region. Nivolumab entered the clinics in 2006, and pembrolizumab as well as atezolizumab followed in 2011 [185]. Subsequently, cemiplimab (Libtayo^®^, Regeneron, Tarrytown, NY, USA), dostarlimab (Jemperli^®^, GlaxoSmithKline, Brentford, UK), toripalimab (Loqtorzi^®^, Coherus Bioscience, Redwood City, CA, USA), and tislelizumab (Tevimbra^®^, BeiGene, Cambridge, MA, USA) have been approved by the FDA as PD-1 binders for cancer treatment. Additionally, two antibodies targeting PD-L1 have also secured approval within the past 10 years. These drugs, durvalumab (Imfinzi^®^, AstraZeneca, Cambridge, UK) and avelumab (Bavencio^®^, EMD Serono, Rockland, MA, USA), also serve to disrupt the binding of PD-1 and PD-L1 for treatment of solid cancers [186].

### 4.4. Rationale for Combining ICI with TαT

Despite the number of approved ICIs and their use in a range of cancers, many tumors do not respond, or only partially respond, to this treatment. To overcome pre-existing or acquired escape mechanisms, ICI could be combined with TαT to render tumors even more T-cell inflamed [187]. Multiple studies have revealed that tumor cells become ‘’immunogenic’’—subject to both recognition by immune cells and immune-mediated killing—upon exposure to various radiation arts [185]. Cells undergoing apoptosis produce many damage-associated molecular patterns (DAMPs) which contribute to radiation-induced immunogenic cell death [188]. DAMPs activate dendritic cells, resulting in improved tumor antigen presentation, stimulation of cytotoxic T lymphocytes, and the release of cytokines and chemokines, all of which help the immune system respond to cancer cells [188,189]. TαT potentially induces significant cell damage and apoptotic cell death and might therefore be expected to enhance the effect of ICI immunotherapy by increasing T cell activation (Figure 4). Finally, because the mechanisms of tumor cell killing are not overlapping, ICI immunotherapy and TαT could combine to attack tumors that do not homogenously express the target of the TαT agent or are poorly accessible to T cells and immunotherapeutics.

### 4.5. Pre-Clinical and Clinical Insights: ICI and TαT Synergy

The journey of combining ICI with TRNT showcases a series of groundbreaking experimental and clinical endeavors that illuminate the immunostimulatory effects of β^−^-particles (Table 3). These initiatives have collectively demonstrated the enhanced efficacy of this combined approach, paving the way for novel cancer treatment strategies. The exploration into this combinatorial treatment began in 2018 with Jaeyeon et al., who reported the effective use of very late antigen-4 (VLA-4)-based TRNT with [^177^Lu]Lu-LLP2A and ICI in the B16-F10 melanoma model. This study provided early evidence that combining TRNT with ICI could significantly improve survival rates in aggressive metastatic melanoma [190]. Building on this, in 2019 Chen et al. tested a novel therapeutic strategy combining anti-PD-L1 immunotherapy with peptide-based TRNT using [^177^Lu]Lu-EB-RGD in a murine colon cancer model. This approach achieved an increase in PD-L1 expression in T cells and enhanced CD8+ T cell infiltration, thus improving local tumor control and overall survival [191]. Kim et al., in 2020, contributed further to the field with a phase I study on [^177^Lu]Lu-DOTA-TATE (Lutathera^®^, Novartis) combined with nivolumab for neuroendocrine tumors of the lung, expanding the potential applications of TαT and ICI combinations in cancers not traditionally seen as highly immunogenic [192]. Another study by Rouanet et al., in 2020, showed that [^131^I]ICF01012 induced immunogenic cell death and improved survival in immunocompetent mice while enhancing immune cell recruitment in the tumor microenvironment. The combination of [^131^I]ICF01012 with ICI such as anti-CTLA-4, anti-PD-1, and anti-PD-L1 not only prolonged survival significantly (compared to TRNT alone) but also showed greater efficacy compared to ICIs alone without added toxicity [193].

Utilizing α-particles, Czernin et al. explored the synergistic effects of [^225^Ac]Ac-PSMA-617 and PD-1 blockade in a mouse model of PCa, finding that these treatments significantly enhanced therapeutic outcomes. While both [^225^Ac]Ac-PSMA-617 and anti-PD-1 antibody treatments individually prolonged time to progression (TTP) and survival, their combination further extended TTP to 47.5 days and increased survival to 51.5 days compared to either treatment alone (anti-PD1 alone 37 days, [^225^Ac]Ac-PSMA-617 32 days). This suggests that combining PSMA-TαT with immunotherapy may be a promising approach for improving disease control in PCa which is per se also only poorly responding to ICI [194]. In 2021, the combination of TRNT using radiolabeled alkylphosphocholine (APC) analog [^90^Y]Y-NM600 and ICI was shown to enhance the CD8+ immune response against T-cell non-Hodgkin lymphoma showed strong therapeutic effects, with 45 to 66% of mice exhibiting complete tumor response and developing tumor-specific T cell memory compared to none with either treatment alone [195]. Recent research has explored new treatment strategies for aggressive cancers by combining TRNT and immune checkpoint blockade (CP). One triple-negative breast cancer (TNBC) study revealed the combination of TCO-BSA labelled with Lu-177 and CP to extend survival and enhance the anti-tumor immune response in a murine model. These studies collectively highlight the promising potential of integrating TRNT with immunotherapy in treating resistant and aggressive cancer types [196]. Similarly, in a documented case report, the patient, having undergone multiple prior treatments, received a combined regimen of [^177^Lu]Lu-PSMA-617 and immunotherapy. This approach effectively mitigated tumor progression in uterine leiomyosarcoma, thereby demonstrating the potential applicability of this strategy across a spectrum of PSMA-positive cancers [197].

Limited instances of preclinical or clinical studies have evaluated the combination of TαT and ICI immunotherapy to further confirm the combined approach to prolong overall survival compared to the respective monotherapies. The following clinical studies exemplify the ongoing exploration into the integration of TαT and immunotherapy for cancer treatment. The clinical trial NCT04946370 investigates the combination of [^225^Ac]Ac-J591 with immune-modulating capabilities of pembrolizumab, further highlighting the innovative approach. To this date, no results were posted yet.

Multiple clinical trials are exploring the synergy between Ra-223 (Xofigo^®^) and various ICIs for advanced cancers. The NCT02814669 trial combined atezolizumab (targeting PD-L1) with Xofigo^®^ for mCRPC. It reported an increase in adverse events, including severe reactions, and failed to demonstrate clinical benefits, raising questions about the effectiveness of this combination for mCRPC. The NCT03996473 trial investigated the use of pembrolizumab (targeting PD-1) with Xofigo^®^ in stage IV NSCLC patients with bone metastases. The outcomes were mostly negative, with the majority of participants showing disease progression; only one achieved stable disease. Due to these disappointing results, the study was terminated early and did not move to phase II. Additional ongoing studies, such as NCT04071236 and NCT04109729, continue to test different ICIs like nivolumab and avelumab alongside Xofigo^®^. These trials remain active and are currently recruiting participants, aiming to uncover potentially improved treatment options for various types of cancer.

### 4.6. Future Perspectives and Challenges

The prevalence of immunologically “cold” tumors, as well as the exhaustion of T cells, upon persistent exposure to tumor antigens creates a need for a combination therapeutic that is efficacious in these tumors and (re)sensitizes tumor cells to the host immune system. TαT is ideally positioned to fulfill this role because it kills tumor cells independently of immune activity and also produces pro-inflammatory cytokines in the TME, generates new antigens, and promotes tumor infiltration and clonal expansion of CD8+ T cells. Preliminary preclinical studies support the hypothesis that TαT and ICI immunotherapy combine additively. Furthermore, the immune response can be further augmented by applying systemic immune-stimulating agents. This provides a rationale for further preclinical and clinical proof-of-concept studies to evaluate the therapeutic potential of TαT combinations with immunotherapies and immune-stimulating agents for specific and durable anti-tumor response.

**Table 3 pharmaceuticals-17-01031-t003:** Overview of most recent and relevant publications on Immunotherapy with Checkpoint Inhibitors (ICI) as monotherapy and in combined treatment with TαT (refer to Section 4). To facilitate the orientation, the table groups references on ICI as a monotherapy first, followed by references on combination of ICI with TRNT. Note: Only the most recent and relevant publications are included in the table to provide readers with a solid overview. More details can be found in Section 4.

Method	Authors	Year	Study	Cancer Type	ICI Agent	TRNT Agent	Main Findings	Ref.
**ICI**	Bonaventura et al.	2019	Review	Various	Various	NA	Cold tumors: A therapeutic challenge for immunotherapy.	[187]
	Sun et al.	2020	Meta-analysis	Various	Anti-PD-1, Anti-PD-L1	NA	Clinical efficacy and safety of anti-PD-1/PD-L1 inhibitors for advanced or metastatic cancer.	[178]
	Arina et al.	2020	Review	Various	Various	NA	Radiotherapy and immunotherapy for cancer: From ‘systemic’ to ‘multisite’.	[185]
	Yap et al.	2021	Review	Various	Various	NA	Development of immunotherapy combination strategies in cancer.	[171]
	Sharma et al.	2023	Review	Various	Various	NA	Immune checkpoint therapy—current perspectives and future directions.	
**ICI & TRNT**	Chen et al.	2019	Pre-clinical	Colon cancer (MC38)	Anti-PD-L1	[^177^Lu]Lu-EB-RGD	Increased PD-L1 expression on T cells, enhanced CD8+ T cell infiltration, and improved local tumor control.	[191]
	Patel et al.	2021	Pre-clinical	Various (B78, 4T1, NXS2, Panc02, B16)	Anti-CTLA-4, Anti-PD-1, Anti-PD-L1	[^131^I]I-CF01012	Enhanced CD8+ immune response.	[195]

## 5. Integration of Cytostatic Chemotherapy (CCT) with Targeted Alpha Therapy (TαT)


**Key Facts:**
Combining CCT with TαT can enhance treatment effectiveness, increasing cytotoxicity due to high-LET radiation, potentially allowing for lower doses of each agent to reduce side effects.Studies show promise in enhancing the efficacy of taxanes through new formulations and combinations with other treatments. Early clinical trials indicate the potential of taxane and TαT combinations in PCa treatment, with ongoing studies aiming to confirm these findings.


**Figure 5 pharmaceuticals-17-01031-f005:**
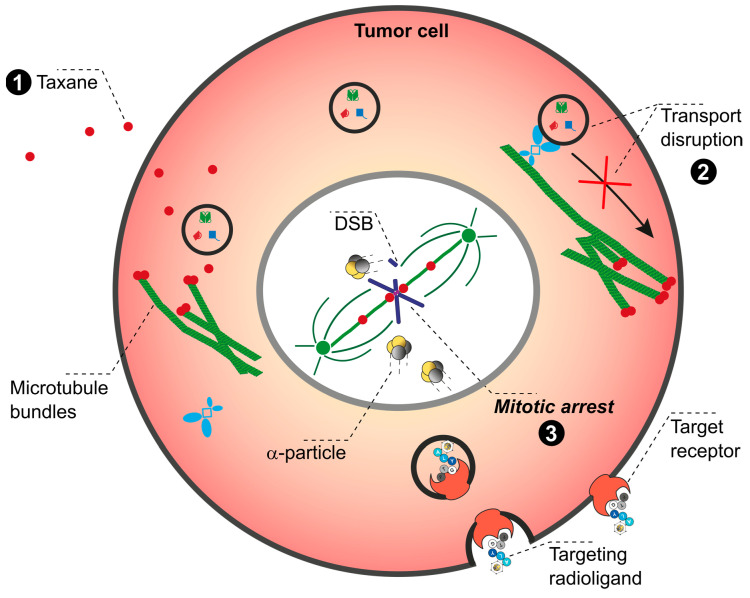
**Graphical representation of systemic CCT and systemic TαT combination therapy**. The illustration depicts single tumor cell under mitotic arrest and transport disruption. **Taxanes (1)** bind to microtubules and significantly increase their stability, leading to **transport disruption (2)**. This has a detrimental effect on the cell by inducing **mitotic arrest (3)**, DNA damage, chromosome segregation issues, and significant disturbance of cellular transport. The combination of TαT and taxane-based chemotherapy—docetaxel and cabazitaxel—is based on two different mechanisms which are potentially mutually reinforcing. TαT induces complex damage to the cellular DNA, which activates DNA repair pathways. Treatment with taxanes, however, disturbs the DNA repair machinery by induction of mitotic arrest and disruption of cellular trafficking. Therefore, the combination of the damage-inducing TαT with a DNA-repair-inhibiting CCT agent could lead to enhanced damage and ultimately cell death of tumor cells. Additionally, cell death induced by TαT is more specific to the tumor cells rather than rapidly-dividing cells—tumor or healthy alike—which are the primary targets of CCT. Furthermore, as both approaches represent systemic treatments, the combination might be highly promising for patients with a rather advanced disease stage and already developing treatment resistance. Note: Due to space constraints, illustrated examples do not cover the full spectrum of possible chemotherapeutic agents and focus exclusively on taxane-based drugs.

### 5.1. Comprehensive Overview of CCT in Cancer Treatment

Chemotherapy is an important and widely used treatment modality in many different cancer types. The use of chemical agents as potential anticancer drugs started in the 1940s with nitrogen mustard and quickly expanded over the following decades, both in terms of the number of chemotherapeutic agents and in terms of their frequency of use, to become one of the leading treatment modalities for cancer. As both the library of chemotherapeutic agents and their target cancer types are vast and the scope of this review is limited, we will mostly focus on PCa and on taxanes as therapeutic agents to illustrate the potential of combining various therapeutic approaches to potentially increase overall treatment efficiency [198,199,200,201,202,203].

In PCa, chemotherapy serves as a first and second line of treatment, usually alongside other treatment modalities [204]. Out of the various cytotoxic agents tested so far, only docetaxel (Taxotere^®^, Sanofi-Aventis, Paris, France; Docefrez^®^, Sun Pharma Global, Mumbai, India; Zytax^®^, Zydus, Ahmedabad, India), and cabazitaxel (Jevtana^®^, Sanofi-Aventis) showed clear overall survival benefits. Both docetaxel and cabazitaxel belong to the taxane family of microtubule-targeting agents (MTAs) [204,205,206]. These drugs contribute to tumor cell killing through multiple mechanisms summarized in Section 5.2.

### 5.2. Mechanistic Insights into Taxane-Induced Cytotoxicity

Taxanes stabilize microtubules against disassembly through binding β-tubulin to counteract the effect of hydrolysis of guanosine-5′-triphosphate (GTP) to guanosine-5′-diphosphate (GDP) and promote the assembly of tubulin dimers into microtubules [207]. This action leads to a disturbance in normal microtubule dynamics which—depending on the cell cycle phase—can lead to various catastrophic effects [208,209].

One of the most studied catastrophic effects is the induction of mitotic arrest. In mitotic cells, taxanes bind to, and stabilize, the mitotic spindle and prevent sister chromatid segregation. This in turn activates the spindle assembly checkpoint, which triggers a mitotic arrest [209]. Prolonged mitotic arrest is detrimental to cells, and it often leads to cell death. Another possible toxic effect of taxanes during mitosis is multipolar spindle formation. This happens at lower taxane concentrations, and while it does not trigger mitotic arrest, it can lead to serious chromosome mis-segregation and subsequent cell death [210]. Although the cell killing effect of taxanes was mainly linked to their mitotic effects in rapidly dividing cells—which describes both in vitro cell cultures and xenografts—the doubling time of human tumor cells is generally much longer than that of cell lines or xenografts [211]. As an example, human PCa cells have an average proliferative doubling time of 19–108 days, much slower than model cells cultured in vitro. Nevertheless, both docetaxel and cabazitaxel are effective against these cells [212]. This discrepancy led to the deeper exploration of taxane-related effects on interphase cells and turned up new, interphase-specific cytotoxic effects.

Recent studies showed that docetaxel and cabazitaxel induce microtubule bundling in interphase cells and cause major disruptions in cellular trafficking. As a result, essential proteins like DNA damage repair proteins or various receptors and transcription factors (e.g., androgen receptor) cannot be properly translocated to the nucleus [213,214,215]. Disrupted intracellular trafficking can severely disrupt normal signal transduction pathways and often leads to apoptotic cell death. Additionally, functional microtubule dynamics are necessary for DNA DSB repair. In interphase, this process happens via NHEJ, which requires a tightly regulated mobile DSB, for which a functional microtubule system is essential [216]. In the absence of proper microtubule dynamics, DSB accumulation or mis-regulated NHEJ can lead to severe genomic instability and cell death.

A lesser-known effect of docetaxel and cabazitaxel, which may also contribute to their efficacy in slower-growing prostate tumor cells, is the promotion of apoptosis in a microtubule-independent manner. The mechanism behind this is not fully understood; however, the observation that these drugs decrease the expression level of an anti-apoptotic protein Bcl-2 is likely relevant [217,218]. In the case of cabazitaxel, this effect is also paired with an increase in the expression of Bax, a pro-apoptotic member of the Bcl-2 family [219]. Together, these changes can lead to mitochondrial outer membrane permeabilization and to an increase in activated caspase-3 levels. Caspase-3 may be responsible for coordinating the apoptosis.

### 5.3. Current Advances and Innovations in CCT

Taxanes such as docetaxel remain a key component of SoC protocols in multiple cancer types [220]. However, they are systemic drugs that do not selectively target cancer cells and consequently have multiple side effects on dividing cells [204,221,222]. These side-effects are partially caused by non-specific cancer targeting, relatively low tumor accumulation, and the requirements for irritating surfactants to improve taxane solubility, which collectively limit the treatment doses, reduce efficacy at the administered dose, and can severely impact the patient’s quality of life [220,221,223]. Therefore, much of the recent work with taxanes has focused on developing new formulations to improve tumor uptake and reduce access to off-target tissue as well as combining them with other chemotherapeutic drugs or even different treatment modalities. Among these efforts, nanoparticle formulations are playing a leading role [224,225]. For example, as part of a cross-over study, the plasma and intra-tumoral pharmacokinetics of CPC634—a nanoparticle entrapping docetaxel—was compared to classically formulated docetaxel [226]. The study showed a significant increase in intracellular docetaxel concentration (461% higher) and a significant decrease in plasma docetaxel levels (91% lower) compared to conventional docetaxel. The study also showed a marked decrease in serious grade 3 neutropenia in patients receiving CPC634 compared to conventional docetaxel; however, other grade 1–2 adverse effects such as skin toxicity and sensory neuropathy incidences were higher in the case of CPC634.

In principle, nanoparticles could facilitate selective tumor targeting by potentially increasing tumor uptake and by eliminating the use of some dose-limiting components of the current formulations [220,223]. Nano formulations are also appealing from a combination therapy perspective, as they could coordinate the selective targeting of tumor cells with both the chemotherapeutical drug and targeted radionuclide. This would potentially allow either or both agents to be administered at decreased dose levels, thereby decreasing the side effects and potentially increasing the patient’s quality of life.

### 5.4. Rationale for Combining CCT with TαT

Although docetaxel and cabazitaxel can be highly effective against cells in interphase and mitosis, tumor cells can acquire resistance against these drugs, decreasing their effectiveness if used alone [214]. Therefore, combining various therapeutic approaches can significantly increase treatment effectiveness [198]. Based on their currently understood mechanisms of cytotoxicity, taxane-based chemotherapy and TαT are good candidates for combination therapy (Figure 5). Indeed, previous preclinical studies indicate that taxanes sensitize various tumors to low-LET radiation. Miyanaga et al. reported that low concentrations of paclitaxel and docetaxel (<1 nM) before irradiation (2–6 Gy X-rays) induced higher amount of DSBs and apoptosis. Zhang et al. observed enhanced cytotoxicity in clonogenic assays when cells were exposed to 20 nM of paclitaxel before radiation treatment (0–8 Gy X-rays) [227,228]. It is plausible that these CCT agents will have the same effect against high-LET radiation, further enhancing the cytotoxicity of TαT. Indeed, the effect may be even stronger using α-emitters because these agents induce DSBs at a higher rate than low-LET, rendering cells even more vulnerable to the impaired NHEJ repair induced by taxanes during interphase. Through improved efficacy, it might be possible to reduce the doses administered of each agent. To wit, the radiation sensitizing effect is evident even at a low taxane concentration [228], which would reduce the chemotherapeutic burden, while the cells would be sensitive to lower α-radiation doses, which would also reduce the radiation burden.

### 5.5. Pre-Clinical and Clinical Insights: CCT and TαT Synergy

So far, research on a combined taxane and TαT therapy in either pre-clinical or clinical settings is at its nascent stage. However, as stated in Section 5.3, there is a considerable effort to improve the efficiency and utility of taxanes in a clinical setting (Table 4).

BIND-014, a PSMA-targeted nanoparticle, has also been developed to improve CCT in mCRPC. In a phase II clinical trial, the safety and efficiency of BIND-014 was tested in 42 mCRPC patients [229]. Although the study did not compare BIND-014 directly to conventional docetaxel treatment, the authors saw some encouraging results. For example, the median radiographic progression-free survival (PFS) was 9.9 months, which was better than the pre-specified favorable outcome for the trial (≥6 months). Additionally, almost a third of the patients showed a ≥50% decrease in PSA levels, while half of the patients with unfavorable circulating tumor cell counts saw a reduction to these values.

Yet another new formulation currently under investigation is albumin-bound paclitaxel (Nab-paclitaxel), a different member of the taxane family. In a recent phase III clinical trial, Nab-paclitaxel was compared to conventional docetaxel in non-small cell lung cancer (NSCLC) patients previously pre-treated with cytotoxic chemotherapy [230]. The study showed slightly increased overall survival (OS; 16.2 vs. 13.6 months) and increased PFS (4.2 vs. 3.4 months) for patients receiving Nab-paclitaxel vs. docetaxel. Interestingly, while conventional docetaxel treatment caused higher incidence of grade 3–4 neutropenia or febrile neutropenia than Nab-paclitaxel, Nab-paclitaxel treatment had a higher incidence of grade 3–4 peripheral sensory neuropathy.

In addition to these comparison studies, Nab-paclitaxel is also being used in various trials in combination with other treatment modalities. These include, but are not limited to, several phase III clinical trials focusing on triple-negative breast cancer (TNBC) or squamous non–small-cell lung cancer (sNSCLC) investigating the efficiency of combining Nab-paclitaxel with anti-PD-1 (pembrolizumab) and anti-PD-L1 (atezolizumab) antibodies [231,232,233]. In TNBC, the combination of Nab-paclitaxel with atezolizumab showed increased PFS and OS, albeit with grade 3–4 side effects occurring more frequently. These side effects seemed to be in line with the ones described in other studies [231]. Another study focusing on pembrolizumab also assessed the effect of combining the antibody with Nab-paclitaxel, paclitaxel, and gemcitabine as chemotherapeutic agents. Although the results were not broken down based on the chemotherapeutic agent used, the PFS and OS were higher in the combined therapy group than in the placebo-pembrolizumab group [232]. In the case of sNSCLC, the phase III trial focused on the combination of pembrolizumab and either carboplatin or paclitaxel, or carboplatin and Nab-paclitaxel. The study found a higher PFS and OS in the combination therapy groups vs. the placebo-chemotherapy group. Results were also comparable between the paclitaxel and Nab-paclitaxel groups.

To our knowledge, research on a combined taxane and TαT therapy has not been published. However, there are a handful of clinical studies of TRNT in combination with taxanes that support this approach. A recent case study highlighted the promise of taxane chemotherapy with targeted radiopharmaceutical therapeutics in a patient with PCa [234]. The patient received an initial course of [^177^Lu]Lu-PSMA-617. After new bone lesions developed, a second sequence was initiated in combination with low doses of paclitaxel (Taxol^®^) to act as a radiation sensitizer. This second course led to remarkable and sustained tumor reduction with only transient, treatable side effects. A recently completed phase I trial (NCT00916123) in 15 men with PCa treated in combination with PSMA-targeting antibody [^177^Lu]Lu-J591 and docetaxel concluded that TRNT could be safely conducted concurrently with standard taxane chemotherapy [235]. In parallel, a recently opened phase I/2 trial (LuCAB, NCT05340374) will assess the safety and efficacy of cabazitaxel in combination with [^177^Lu]Lu-PSMA-617 in PCa patients [236].

### 5.6. Future Perspectives and Challenges

Multiple recent and ongoing clinical trials focus on testing new taxane formulations and combination therapies of taxanes with other treatment modalities. Many of these trials show that both of these approaches can offer increased survival benefits to patients. Given the rapid development of new taxane formulations, the safety of taxanes in combination with TRNT and the potentially additive interplay between TαT and taxanes, such a combination therapy, holds real potential to improve outcomes in cancer.

**Table 4 pharmaceuticals-17-01031-t004:** Overview of most recent and relevant publications on Cytostatic Chemotherapy (CCT) as monotherapy and in combined treatment with TαT (refer to Section 5). To facilitate the orientation, the table groups references on CCT as a monotherapy first, followed by references on combination of CCT with TRNT. As mentioned in Section 5.5, there are currently no pre-clinical or clinical studies on combination of CCT and TαT. However, most TαT patients went through CCT before getting assigned to TαT. Note: Only the most recent and relevant publications are included in the table to provide readers with a solid overview. More details can be found in Section 5.

Method	Authors	Year	Study	Cancer Type	CCT Agent	TRNT Agent	Main Findings	Ref.
**CCT**	Xu et al.	2022	Pre-clinical	PCa	Cabazitaxel	NA	Cabazitaxel suppresses the proliferation and promotes apoptosis and radiosensitivity: suppression of PI3K/AKT pathway.	[218]
	Schmid et al.	2018	Clinical	Breast cancer (TNBC)	Nab-paclitaxel	NA	Atezolizumab and nab-paclitaxel combined prolonged progression-free survival in both the intention-to-treat population and PD-L1–positive subgroup.	[231]
	Paz-Ares et al.	2018	Clinical	Lung cancer (NSCLC)	Paclitaxel/Nab-paclitaxel	NA	Combination of pembrolizumab to chemotherapy with carboplatin plus paclitaxel/nab-paclitaxel lead to significantly longer overall survival and progression-free survival than chemotherapy alone.	[233]
	Yoneshima et al.	2021	Clinical	Lung cancer (NSCLC)	Nab-paclitaxel/Docetaxel	NA	Nab-paclitaxel was noninferior to docetaxel in terms of OS.	[230]
	Cortes et al.	2022	Clinical	Breast cancer (TNBC)	Paclitaxel/Gemcitabine–carboplatin	NA	Patients with high PD-L1 expression benefit from combination of pembrolizumab and chemotherapy: significantly longer overall survival than chemotherapy alone.	[232]
**CCT & TRNT**	Maharaj et al.	2021	Clinical	mCRPC	Docetaxel	[^177^Lu]Lu-PSMA-617	Low-dose docetaxel as radiosensitizer with [^177^Lu]Lu-PSMA-617 showed good response; no tumor resistance.	[234]
	Batra et al.	2020	Clinical	mCRPC	Docetaxel	[^177^Lu]Lu-J591	Combination of [^177^Lu]Lu-J591 (single fractionated cycle) with docetaxel was well tolerated.	[235]
	Kostos et al.	2023	Clinical	mCRPC	Cabazitaxel	[^177^Lu]Lu-PSMA-617	Evaluation of cabazitaxel in combination with [^177^Lu]Lu-PSMA-617 (LuCAB) to target micrometases.	[236] NCT05340374

## 6. Integration of Brachytherapy (BT) with Targeted Alpha Therapy (TαT)


**Key Facts:**
Combining BT with TαT aims to enhance tumor cell killing by addressing regions with insufficient TαT dose, known as ‘cold spots’, leading to a uniform tumor control while sparing healthy tissues.New methods like diffusing α-emitters radiation therapy (DαRT) are being tested. Combining BT and TαT might enhance anti-tumor immune responses and improve therapeutic outcomes.


**Figure 6 pharmaceuticals-17-01031-f006:**
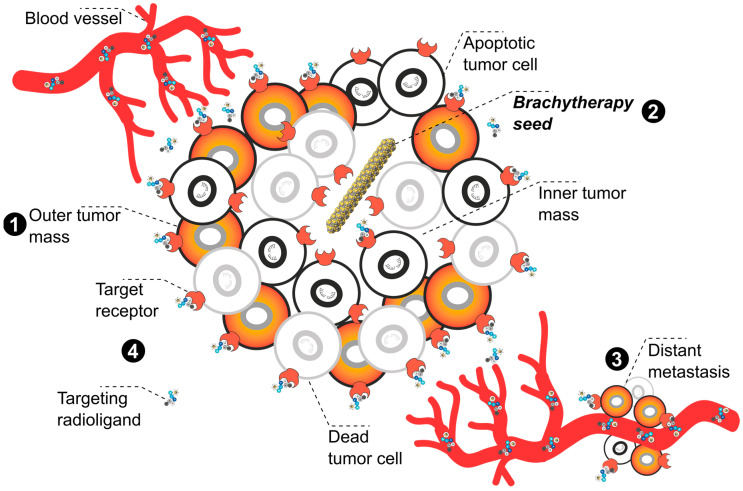
**Graphical representation of the local BT and systemic TαT combination therapy**. The illustration depicts **tumor mass (1)** with the **brachytherapy seed (2)** inserted in the middle of the tumor. The primary tumor and **distant metastases (3)** along blood vessels are treated by **targeting radioligand** docking on target receptors **(4)**. The inserted seed releases energy from radioactive decay along its path, efficiently eradicating the inner tumor mass and preventing relapse in tumors with ‘cold spots’ that lack target expression for TαT. Tumors expressing TαT targets enable the treatment of not only primary tumors but also metastatic lesions. Combination therapy with BT and TαT relies on the killing efficiency of various radionuclides. Specifically, the killing efficiency would be enhanced by attacking the tumor mass from two sites: cavital and systemic. The insertion of a radioactive seed inhibits the relapse in tumors that do not express specific targets for TαT but might have non-irradiated ‘cold spots’ in the tumor periphery that eventually lead to a relapse. In tumors that exhibit appropriate targets for TαT, this approach ensures comprehensive coverage of the tumor’s ‘cold spots’. When used in conjunction with BT, TαT can potentially lead to the complete eradication of both the primary tumor and any metastatic lesions.

### 6.1. Comprehensive Overview of BT in Cancer Treatment

Brachytherapy (BT) is a specific form of delivering radiation through sealed radioactive sources. This therapy is most commonly used to treat head and neck, breast, cervical, and prostate cancers. The radiation sources are implanted in the tumor via a catheter or, occasionally, intracavitary, intraluminal, or interstitial (needle) applicators. The efficacy of BT is based on the direct radiation of the tumor cells. In this respect, it is functionally analogous to TRNT and TαT [237].

### 6.2. Radiobiological Mechanisms Underlying BT

DNA is the primary cellular target of BT. If unrepaired, radiation-induced DNA damage leads to cell cycle redistribution, microenvironmental changes, and cell death. The therapeutic index of irradiation relies on the differential response between tumors and normal tissue, as tumor cells have a lower DNA repair capability compared to normal tissue cells. However, radiation-induced changes in organs at risk may lead to acute side effects and long-term functional sequelae. Therefore, the ideal technique should be able to deliver therapeutic doses to the tumor with doses as low as possible to the organs at risk [238].

### 6.3. Current Advances and Innovations in BT

Brachytherapy is an effective tool for obtaining high tumor doses while minimizing the exposure to organs at risk, as documented by dosimetry studies. The total dose can be delivered by one of three methods: (1) continuous low-dose-rate (LDR) irradiation, (2) pulse-dose-rate irradiation, whereby low-intensity pulses are repeated every hour for up to a few days, or (3) a few fractions that deliver high doses constantly (high-dose-rate [HDR] irradiation) [239]. Iridium-192 (Ir-192, τ_1/2_ = 73.8 d, 95.6% β^−^ decay), cobalt-60 (Co-60, τ_1/2_ = 5.27 years, 100% β^−^ decay), iodine-125 (I-125, τ_1/2_ = 59.4 d, 21 Auger electrons per decay), and palladium-103 (Pd-103, τ_1/2_ = 16.99 d, >24 Auger electrons per decay) are the radioisotopes that are commonly applied in modern brachytherapy [240]. The clinical use of brachytherapy has notably reduced over time. Among the reasons for its waning popularity are the lack of sophisticated training programs for physicians for placement of the implants and the invasive nature of the implantation, which requires a short procedure in the operating room to place the sources or the catheters and can cause pain when the catheter is removed. Additionally, any inaccurate positioning of the radiation sources may decrease the therapeutic index by exposing the tumor to underdosage or neighboring tissue to excessive dose [241].

### 6.4. Rationale for Combining BT with TαT

The conceptual basis for integrating BT and TαT lies in harnessing the cytotoxic capabilities of multiple radiation sources delivered through distinct methodologies. Although there are currently no clinical or pre-clinical studies directly investigating the combined effects of BT and TαT, parallels drawn from existing radiation combination strategies, such as those involving EBRT and TRNT, suggest potential efficacy. This analogy supports the hypothesis that combining BT and TαT could prove successful in enhancing therapeutic outcomes by exploiting synergistic effects of different radiation modalities to achieve more comprehensive tumor control.

Systemic administration of the TαT agent will cause it to kill cells expressing the desired target, but, through heterogenous target expression and vascularization or total lesion volume, not all regions of the tumor will receive a lethal dose. These ‘cold spots’ could be the genesis of relapse. When optimally combined, BT could be used to kill the tumor regions that receive insufficient doses from TαT therapy. By addressing these ‘cold spots’, this combination therapy could lead to more uniform tumor cell killing and lasting tumor control, all while sparing healthy surrounding tissues (Figure 6).

### 6.5. Pre-Clinical and Clinical Insights: BT and TαT Synergy

BT is commonly used alone or alongside conventional CCT, surgery, or EBRT in clinical settings. While BT has been explored in combination therapies for PCa, no studies have yet investigated its potential synergies with TαT (Table 5). Patients with low and intermediate-risk disease are eligible for LDR BT, which requires the permanent insertion of radioactive I-125 seeds (or, less frequently, Pd-103 seeds) into the prostate. The procedure offers a number of benefits over alternatives such as radical prostatectomy, including being less intrusive and associated with a lower risk of deficits in long-term quality of life. It can be performed as an outpatient procedure under spinal anesthesia, with a good recovery and a lower risk of dribbling [242]. The method of delivering prostate HDR BT over a few sessions involves introducing catheters inside the gland while using high-activity radioactive sources of Ir-192. In this setting, BT may be used in combination with EBRT to reduce the dose requirement. As evidence for this hypothesis, a randomized clinical trial in patients with localized or locally advanced PCa comparing the survival and toxicity for HDR BT in combination with EBRT and EBRT alone found that treatment with HDR BT before EBRT reduced the required dose of EBRT to achieve a comparable survival outcome at significantly lower side effects [243]. To further highlight the benefits of combined BT and EBRT, patients that underwent the combined therapy required a shorter duration of androgen deprivation therapy (ADT) treatment [244].

After guidelines for BT and radiation safety were issued in 2003, the clinical use of permanent BT using I-125 seeds to treat PCa has rapidly grown in Japan. Studies in Japanese patients confirmed that the clinical outcomes of PCa patients receiving combined BT and EBRT were significantly improved. Notably, patients who underwent combined BT and EBRT without adjuvant ADT had a higher probability of improved survival [245]. In the largest trial to evaluate BT and EBRT, the ASCENDE-RT (Androgen Suppression Combined with Elective Nodal and Dose Escalated Radiation Therapy) study, 398 patients with intermediate-risk and high-risk cancer were enrolled. That trial showed that the probability of biochemical failure at a median follow-up of 6.5 years was twice as high among patients who received an EBRT boost compared with those who received a BT boost [246]. Finally, Oshikane et al. recently reported that combined HDR BT and EBRT led to a significantly improved therapeutic outcome compared to EBRT alone in PCa patients receiving ADT [247].

A new iteration of BT, diffusing α-emitters radiation therapy (DαRT), is now being evaluated in clinical trials as a potential treatment for solid tumors [248]. DαRT utilizes interstitial seeds of Ra-224 at 10–1000 kBq activity levels which emit short-lived α-particles that disperse throughout the tumor via diffusion. The administration of DαRT treatment, which ostensibly combines BT and TαT in a single agent, elicits specific and systemic antitumor immune responses, ultimately activating the immune system against the tumor. The pro-immunogenic TME is characterized by an augmented infiltration of T lymphocytes and the secretion of granzyme B.

### 6.6. Future Perspectives and Challenges

BT, alone or as part of a combination therapy approach, has historically been an important tool for disease control, curative therapy, and palliative care in a small number of cancers. We have highlighted the combination of HDR BT and EBRT as a particularly effective therapeutic strategy in these cancers and suggest that TRNT, and in particular TαT, could potentially act in synergy with HDR BT or BT to specifically target the tumor and reduce side effects. The benefit of this strategy over existing combinations is the ability to target the tumor mass from its core (through BT) and from its periphery and metastatic sites (through TαT). This combination could lead to more complete tumor cell killing while reducing off-target effects, resulting in a new, safe, and highly potent approach for cancer treatment. The recent emergence of DαRT provides further insight into how BT and TαT could be effectively combined and represents an exciting avenue for study.

**Table 5 pharmaceuticals-17-01031-t005:** Overview of most recent and relevant publications on Brachytherapy (BT) as monotherapy and in combined treatment with TαT (refer to Section 6). To facilitate the orientation, the table groups references on BT as a monotherapy first, followed by references on combination of BT with EBRT. As mentioned in Section 5.5, there are currently no pre-clinical or clinical studies on combination of BT and TαT. Note: Only the most recent and relevant publications are included in the table to provide readers with a solid overview. More details can be found in Section 5.

Method	Authors	Year	Study	Cancer Type	Main Findings	Ref.
**BT**	Annede et al.	2020	Review	Various	Foundations and new insights in radiobiology modelling for brachytherapy effects.	[238]
	Fonseca et al.	2020	Review	Various	Requirements and future directions for in vivo dosimetry in brachytherapy.	[239]
	Ito et al.	2018	Clinical	PCa	Analysis of survival outcomes for permanent I-125 seed implantation in PCa.	[245]
	Arazi et al.	2020	Modeling Study	Various Solid Tumors	Modelling of macroscopic α-particle dose for Ra-224 in DαRT.	[248]
	Tamihardja et al.	2022	Clinical	PCa	Comparison between Ir-192 and Co-60 sources in HDR brachytherapy for PCa.	[240]
	Morris et al.	2017	Clinical	PCa	Patients with LDR brachytherapy were twice as likely to be free of biochemical failure at a median follow-up in comparison to those receiving dose-escalated EBRT.	[246]
**BT & EBRT**	Mori et al.	2021	Clinical	PCa	High risk patients benefit from trimodal therapy with HDR brachytherapy, hypofractionated EBRT, and ADT.	[244]
	Oshikane et al.	2021	Clinical	PCa	HDR brachytherapy boost combined with EBRT has significantly higher biochemical-free survival rate than EBRT alone in high-risk PCa.	[247]

## 7. Outlook

This manuscript presents an overview of TαT to treat cancer and highlights a wide array of established therapeutic strategies that could potentially be combined with TαT for improved outcomes. While some of these combination approaches are undergoing preliminary (pre)clinical evaluation in a range of cancers, others represent untapped potential whose synergistic or additive therapeutic outcomes we hypothesize based on the mechanisms of action of the respective monotherapies. As much as combination therapy with TαT could lead to significantly improved treatment outcomes, it is equally important to acknowledge the biological, logistical, and economic obstacles to the translation of these approaches. Additionally, combined treatment requires the willingness of different medical disciplines to join forces for the greater benefit of the patient. However, the exchange and collaboration between different disciplines is often challenging. This is also related to the economics of the current health care system, which would require different disciplines to share the income for different combined treatments. 

Furthermore, we would like to briefly discuss the current status of radiopharmaceutical production as well, with a particular focus on the prevailing shortage of α-emitting radionuclides, infrastructure considerations, and the optimal treatment sequencing required to incorporate TαT into existing treatment paradigms. It is our expectation that breakthroughs in production methodologies, purification strategies, and radioisotope distribution networks will soon be achieved, thereby facilitating the expansion of TαT to larger numbers of patients and medical centers. With concurrent investment in infrastructure, including increased capacity of treatment centers, education programs for patients and staff to be exposed to new sources of radiation, and radioactive waste management solutions, TαT combination therapy could be administered in a cost-effective and resource-efficient manner. An essential aspect that needs to be investigated is the optimal sequencing of therapies. It is crucial to determine whether a single dose or a fractionated approach would yield superior outcomes. The starting points for these studies should be based on the mechanism of action of TαT, its side effect profile, and the potential synergies it may have with other treatment modalities. 

Finally, the exploration of triple combination therapies involving TαT opens up intriguing possibilities to further improve treatment outcomes and reduce side effects. For example, some cancers might benefit from EBRT + ICI + TαT to eradicate the primary tumor and metastases through tumor cell irradiation and immune system activation. By examining the synergistic effects and additive benefits of multiple treatment modalities, new treatment cocktails could be devised to truly cure some types of advanced cancers. The successful implementation of such complex strategies would require coordinated efforts from diverse scientific and medical specialties. Nevertheless, the rapid progress of TαT from bench to bedside should foster these collaborations and give us great optimism for future progress in the field.

## 8. Conclusions

In conclusion, this comprehensive review illuminates the foundational principles of five distinct therapeutic strategies as monotherapies and as potential combinations with TαT. The combination of EBRT, BT, CCT, or ICI with TαT all represent feasible treatment strategies that could ultimately be tailored based on each patient’s disease and prior treatment history. Strategic combination of these complementary, but unique, therapeutic approaches on an individualized basis may well dramatically improve the prognosis for advanced-stage cancer patients through robust and durable disease control and improvements in the patients’ quality of life. The collaborative efforts of clinicians, researchers, and practitioners in integrating these therapies demonstrate a commitment to advancing cancer care beyond conventional boundaries. This review sets the stage for future exploration and encourages continued research and clinical innovation aimed at expanding and defining the scope of combined therapeutic interventions for the benefit of cancer patients worldwide.

## Figures and Tables

**Figure 1 pharmaceuticals-17-01031-f001:**
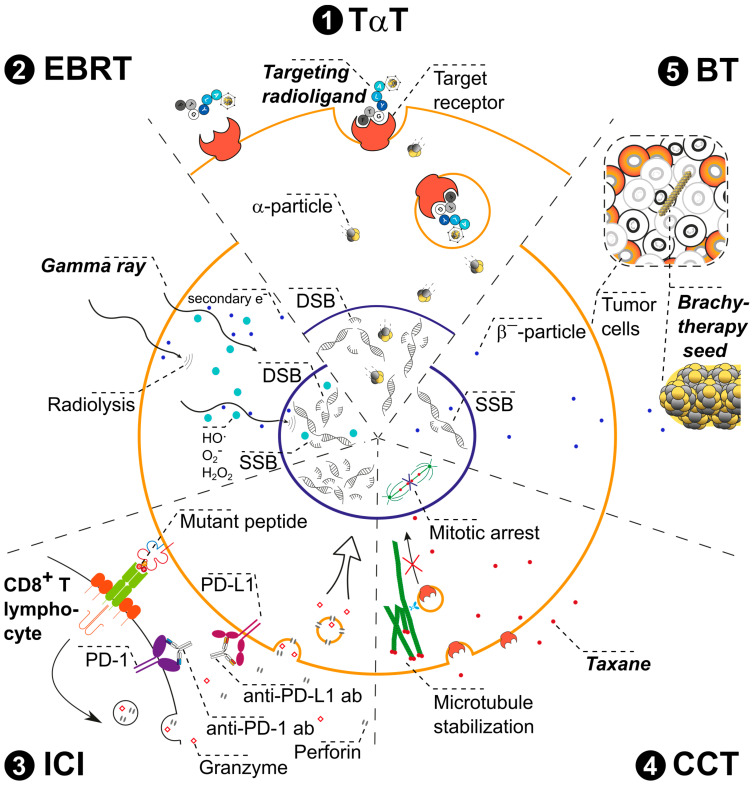
**General overview of multiple local and systemic treatment strategies which could be beneficially combined with TαT**. The upper middle section depicts TαT and its mechanism of action, while the remaining sections illustrate the four pillars of cancer therapies proposed to form a beneficial combination with TαT and their individual mechanism(s) of action. The orange circle represents a cell membrane and the blue circle a cell nucleus. Counterclockwise from top: **(1) Targeted alpha therapy (TαT)** employs ligands carrying α-emitting radionuclides which bind to cancer cells expressing a specific target. The high-LET radiation is responsible for a systemic cell killing via direct and indirect DNA damage. **(2) External beam radiation therapy (EBRT)** is based on the use of ionizing radiation from an external source. The beam facilitates cancer cell killing via direct and indirect DNA and protein damage, resulting from the generation of reactive oxygen species (ROS). **(3) Immune checkpoint inhibitor (ICI)**-based therapies facilitate cancer cell killing by inhibiting immune response escape mechanism, e.g. via the programmed cell death protein 1/programmed death-ligand 1 (PD-1/PD-L1) system. **(4) Cytostatic chemotherapy (CCT)** relies on the use of cytostatic drugs, like taxanes, which induce mitotic arrest and interfere with other microtubule-related functions such as cellular trafficking. **(5) Brachytherapy (BT)** is based on the use of semi-solid therapeutic “seeds”, bearing various radionuclides. The low- or high-LET radiation is responsible for a localized cell killing via direct and indirect DNA damage. Note: Due to space constraints, illustrated examples do not cover the full spectrum of possible vectors, mechanisms of action, and applications, but predominantly focus on the main approaches discussed in this review.

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
