# Peer review of "Future Treatment Strategies for Cancer Patients Combining Targeted Alpha Therapy with Pillars of Cancer Treatment: External Beam Radiation Therapy, Checkpoint Inhibition Immunotherapy, Cytostatic Chemotherapy, and Brachytherapy"

_pharmaceuticals, 2024, doi:10.3390/ph17081031_

Round 1
Reviewer 1 Report
Comments and Suggestions for Authors
This review-type article provided an overview of four therapeutic strategies. It is a long paper that requires significant revision to enhance its quality and clarity. Please refer to my comments as follows.
Comment 1. Refer to the latest journal’s template. The submitted paper is using a template from 2023.
Comment 2. Abstract:
(a) The authors mentioned that “This review aims to provide an overview on four therapeutic strategies”; however, this does not fully align with the paper title.
(b) In addition, the last sentence mentioned another aim, which needs to be matched with the paper title.
(c) The research benefits and implications of this review-type article should be summarized.
Comment 3. More than 230 references are cited. The authors are suggested to update the content with the latest references (mainly recent 5-year; some within 6-10 years) to reflect the latest developments in the research area.
Comment 4. Formal queries should be made using search engines like Web of Science and Scopus to ensure the authors obtain all the latest relevant works.
Comment 5. Tables must be added to ensure a good summary and comparison between works. A review-type article presents only figures, and written text does not facilitate delivering insights and take-away messages.
Comment 6. Section 1 Introduction:
(a) The authors should provide the latest statistics or surveys to justify the importance of the research topic.
(b) Add a paragraph to summarize the research contributions of the paper, preferably in point form for clarity.
Comment 7. Apply this comment to all figures, where applicable:
(a) Enhance the resolutions of all figures. Enlarge the Word file to confirm that no content is blurred.
(b) Ensure in-text citations are given if any figures are referenced to any materials.
Comment 8. Section 2 TαT:
(a) The heading is too simple and confusing with the heading of Subsection 2.1, “TαT Description”.
(b) The heading of Subsection 2.3, “Biology”, is too simple.
(c) As a review-type article, having subsection 2.4 “… State-of-the-Art” is not appropriate.
Comment 9. The rest of the sections follow a similar structure as Section 2, which is not appropriate. As a review-type article, it is expected that the authors provide a clear summary of the existing works with concise comparisons. Particularly, the research challenges and implications should be shared.
There are typos and improper use of English. The organization and structure of the paper should also be improved.
Reviewer 2 Report
Comments and Suggestions for Authors
The authors in the manuscript entitled "Future Focus on Targeted Alpha Therapies in Combination 2 with Other Treatment Strategies" collected and assimilated the literature in a very logical pattern. Introduction section represented bases of the radio-activity/-therapy in addition to necessary background of the subject. The article, reflected by other sections, showed both; the primary and advanced information of the field which could be beneficial for the nuclear medicine/radiopharmaceutical students and readers. The authors have covered maximum reported literature. I my view the review is worth taking. But I suggest authors to address few comments, as follow:
1. Please mark the different parts/stages with numeric numbers of alphabets for explaining in figure caption
2. References are relevant; a good study based on Ac(225)-PSMA should be cited in this review as well,
Rasheed R, Usmani S, Naqvi SAR, Alkandari F, Marafi F. Alpha Therapy with 225Actinium Labeled Prostate Specific Membrane Antigen: Reporting New Photopeak of 78 Kilo-electron Volts for Better Image Statistics. Indian J Nucl Med. 2019 Jan-Mar;34(1):76-77. doi: 10.4103/ijnm.IJNM_115_18. PMID: 30713391; PMCID: PMC6352636.Alpha
Comments on the Quality of English Language
English language used is fine
Reviewer 3 Report
Comments and Suggestions for Authors
The review paper "Future Focus on Targeted Alpha Therapies in Combination with Other Treatment Strategies" is extensive and well-researched. It provides vital insights into the possibilities of combining TαT with other therapeutic tactics. This paper reviews several therapy techniques (EBRT, BT, CCT, ICI) and their possible combination with Targeted Alpha Therapy (TαT). This comprehensive approach helps readers grasp the broader panorama of cancer treatment. The paper is thoroughly researched, citing various studies and clinical trials. Emphasizing the ability of combination medicines to overcome resistance and improve therapeutic efficacy tackles an important issue in cancer. Given the growing interest in personalized medicine, this focus is both relevant and timely. However, various factors must be considered before publication.
- The review is 45 pages long and dense. While extensive, the detailed content may be daunting to those who are not experts in the topic. Summarizing key information and presenting clear, short takeaways may increase reading.
- Some sections, particularly the historical overview, are very detailed, while others, like the discussion on combination strategies, could be expanded. Please try to balance the depth of coverage across different sections to enhance the overall coherence of the review.
- While the review aims to ignite ideas for further optimization of individualized cancer treatment, it could offer more specific future directions and identify potential challenges in the implementation of TαT. Addressing these aspects would provide a more rounded perspective.
Minor points
- Please check that the images are cited in the text, there is an inconsistency in this aspect.
- Some sections do not include references. Why? See, in particular, Section 2.3. Biology
Round 2
Reviewer 1 Report
Comments and Suggestions for Authors
The authors have enhanced the quality of the paper. There are some minor comments to be further considered:
Follow-up Comment 1: In the first column “Combination” of Tables 1 and 2, it causes confusion that this column includes elements, “TalphaT alone” and “EBRT alone”. In addition, is there missing content in “with TRNT”, i.e., missing content before “with”?
Follow-up Comment 2: Add the research implications in the conclusion.
Follow-up Comment 3: The format of references in the list of references is not appropriate. Please ensure to follow the journal’s template.
Reviewer 3 Report
Comments and Suggestions for Authors
The authors have responded to my comments and significantly improved the article. I have no further comments.